# Efficient and Stable Fully Dynamic Facility Location

**Sayan Bhattacharya**
Department of Computer Science
University of Warwick
Coventry, CV47AL, United Kingdom
s.bhattacharya@warwick.ac.uk

**Silvio Lattanzi**
Google Research
silviol@google.com

**Nikos Parotsidis**
Google Research
nikosp@google.com

## Abstract

We consider the classic facility location problem in fully dynamic data streams, where elements can be both inserted and deleted. In this problem, one is interested in maintaining a stable and high quality solution throughout the data stream while using only little time per update (insertion or deletion). We study the problem and provide the first algorithm that at the same time maintains a constant approximation and incurs polylogarithmic amortized recourse per update. We complement our theoretical results with an experimental analysis showing the practical efficiency of our method.

## 1 Introduction

Clustering is a fundamental problem in unsupervised learning with many practical applications in community detection, spam detection, image segmentation and many others. A central problem in this space with a long and rich history in computer science and operations research Korte and Vygen [2018] is the facility location problem(which can also be seen as the Lagrangian relaxation of the classic k–median clustering). Due to its scientific importance and practical applications the problem has been studied extensively and several approximation algorithms are known for the problem Guha and Khuller [1999], Jain et al. [2003], Li [2013]. In addition, the problem has also been extensively studied in various computational model as the streaming model Indyk [2004], Lammersen and Sohler [2008], Czumaj et al. [2013], the online model Meyerson [2001], Fotakis [2008] and dynamic algorithm model Cygan et al. [2018], Goranci et al. [2018], Cohen-Addad et al. [2019], Guo et al. [2020] and many more.

Real world applications nowadays often process evolving data-sets, i.e. social networks continuously evolve in time, videos and pictures are constantly uploaded and taken down from media platforms, news article and blog posted are uploaded or taken down and so on so for. For this reason, it is important to design algorithms that are able to maintain a stable and high quality solution and that at the same time can process updates efficiently. As a consequence, the dynamic and online model have been extensively studied and several interesting results are known for classic learning problems Lattanzi and Vassilvitskii [2017], Chan et al. [2018], Cohen-Addad et al. [2019], Jaghargh et al. [2019], Lattanzi et al. [2020], Fichtenberger et al. [2021], Guo et al. [2021]. In this paper, we extend this line of work by studying the classic facility location problem in the fully dynamic setting.

**Problem definition.** The input to our problem consists of a collection $F$ of *facilities* and a collection $D$ of *clients* in a general metric space. For all $i \in F, j \in D$, we let $d_{ij}$ denote the distance between facility $i$ and client $j$[1]. These distances satisfy triangle inequality. Let $f_i$ denote the *opening cost* of facility $i \in F$. Our goal is to open a subset of facilities $F' \subseteq F$ and then connect every client $j \in D$ to some open facility $i_j \in F'$, so as to minimise the objective $\sum_{i \in F'} f_i + \sum_{j \in D} d_{i_j,j}$. The first sum

---

[1]For simplicity, we assume that all the distances between facility $i$ and client $j$ are polynomially bounded.

in the objective denotes the sum of the opening costs of the facilities in $F'$, whereas the second sum denotes the sum of the connection costs for the clients. Throughout the rest of this paper, we will let $m = |F|$ and $n = |D|$ respectively denote the number of facilities and clients.

We consider a *dynamic setting*, where the input to the problem keeps changing via a sequence of updates. Each update inserts or deletes a client in the metric space. Throughout this sequence of updates, we wish to maintain an approximately optimal solution to the current input instance with small *recourse*. The recourse of the algorithm is defined as follows. Whenever we open a new facility or close an already open facility, we incur one unit of *facility-recourse*. Similarly, whenever we reassign a client from one facility to another, we incur one unit of *client-recourse*. The recourse of the algorithm after an update is the sum of facility-recourse and client-recourse. Note that the recourse of an algorithm naturally captures the consistency of the solution throughout the input updates. So having small recourse is particularly important for real world application where a clustering is served to the user or used in a machine learning pipeline Lattanzi and Vassilvitskii [2017]. Our main result in this paper is summarized in the theorem below.

**Theorem 1.** *There is a deterministic $O(1)$-approximation algorithm for fully dynamic facility location problem that has $O(\log m)$ amortized recourse, under the natural assumption that the distances between facilities and clients and the facility opening-costs are bounded by some polynomial in $m$.*

**Our Technique.** To the best of our knowledge, this is the first algorithm for fully dynamic facility location that can handle non-uniform facility costs and maintains a $O(1)$ approximation with polylogarithmic recourse per update. To obtain our result, we consider a *relaxed* version of the natural greedy algorithm in the static setting. This relaxed greedy algorithm proceeds in iterations. In each iteration the algorithm picks a *cluster*, which specifies a set of currently unassigned clients and the facility they will get assigned to, that has approximately minimum average marginal cost. The algorithm stops when every client gets assigned to some facility. In the dynamic setting, we ensure that the solution maintained by our algorithm always corresponds to an output of the relaxed greedy algorithm (assuming the relaxed-greedy algorithm receives the current input instance). To be a bit more precise, we formulate a set of invariants which capture the workings of the relaxed greedy algorithm, and fix the invariants using a simple heuristic whenever one or more of them get violated in the dynamic setting. The approximation guarantee now follows from the observation that the relaxed greedy algorithm returns a $O(1)$ approximation in the static setting. The recourse bound, on the other hand, follows from a careful token-based argument that is inspired by the analysis of a fully dynamic greedy algorithm for minimum set cover Gupta et al. [2017]. In addition, using standard data structures it is easy to show that the amortized *update time* of our algorithm is $\tilde{O}(m)$. We note that this update time is near-optimum, as each arriving client needs to reveal its distances to the facilities, which also holds for the uniform facility opening cost as pointed out in Cohen-Addad et al. [2019][2].

We also complement our theoretical analysis with an in-depth study of the performance of our algorithm showing that our algorithm is very efficient and effective in practice.

**Related Works.** Facility location and dynamic algorithms have been extensively studied in the algorithm and machine learning literature. Here we give an overview of closely related results.

*Facility Location.* The classic offline metric facility location problem has been extensively studied (see Korte and Vygen [2018]). The best-known approximation for the problem algorithm gives a 1.488 approximation Li [2013]. The problem is known to be NP-hard, and it cannot be approximated within a factor of 1.463 unless **NP** $\subseteq$ **DTIME**$(n^{\log \log(n)})$ Guha and Khuller [1999].

*Dynamic algorithm for clustering and facility location.* Recently several dynamic algorithms have been introduced for clustering and facility location problems Lattanzi and Vassilvitskii [2017], Chan et al. [2018], Cygan et al. [2018], Goranci et al. [2018], Cohen-Addad et al. [2019], Henzinger et al. [2020], Guo et al. [2020], Goranci et al. [2021], Fichtenberger et al. [2021]. The recourse for clustering problems was studied in Lattanzi and Vassilvitskii [2017], then several papers followed up with interesting results on recourse and dynamic algorithms Chan et al. [2018], Cohen-Addad et al. [2019], Henzinger et al. [2020], Fichtenberger et al. [2021].

Efficient algorithm for the fully-dynamic facility location problem are known either when the doubling dimension of the problem is bounded Goranci et al. [2018, 2021] or when the facility have uniform

---

[2]In some works, some assumptions on the form of the input are make, which allow avoiding an $\Omega(m)$ factor, see e.g. Guo et al. [2020] where each arriving client reports its nearest facility uppon arrival.

cost Cygan et al. [2018], Cohen-Addad et al. [2019]. When the facilities can have arbitrary non-uniform costs in a general metric space, prior to our work the best known fully dynamic algorithm achieved a $O(\log |F|)$ approximation with polylogarithmic recourse Guo et al. [2020].

## 2 Our Dynamic Algorithm

In Section 2.1, we present an intuitive overview of our algorithm. We formally describe our algorithm in Section 2.2 and Section 2.3, and analyze its recourse and approximation guarantee in Section 2.4.

### 2.1 An Informal Overview of Our Algorithm

We start by recalling a natural greedy algorithm for the facility location problem in the *static setting*. This (static) algorithm is known to return a constant approximation for the problem Jain et al. [2003]. The algorithm proceeds in *rounds*. Throughout the duration of the algorithm, let $F^* \subseteq F$ and $D^* \subseteq D$ respectively denoted the set of opened facilities and the set of clients that are already assigned to some open facility. Initially, before the start of the first round, we set $F^* \leftarrow \emptyset$ and $D^* \leftarrow \emptyset$. The next paragraph describes what happens during a given round of this algorithm.

Say that a *cluster* $C$ is an ordered pair $(i, A)$, where $i \in F$ and $A \subseteq D$. A cluster is either *satellite* or *critical*. If a cluster $C = (i, A)$ is satellite (resp. critical), then its *cost* is given by $cost(C) := \sum_{j \in A} d_{ij}$ (resp. $cost(C) := f_i + \sum_{j \in A} d_{ij}$). Thus, the only difference between an satellite cluster and a critical cluster is that the cost of the former (resp. latter) does *not* (resp. does) include the opening cost of the concerned facility. The *average cost* of a cluster $C = (i, A)$ is defined as $cost_{avg}(C) := cost(C)/|A|$. Let $\mathcal{C}_{ord}(F^*, D^*)$ be the set of all satellite clusters $C = (i, A)$ with $i \in F^*$ and $A \subseteq D \setminus D^*$, and let $\mathcal{C}_{crit}(F^*, D^*)$ be the set of all critical clusters $C = (i, A)$ with $i \in F \setminus F^*$ and $A \subseteq D \setminus D^*$. In the current round, we pick a cluster $C = (i, A) \in \mathcal{C}_{crit}(F^*, D^*) \cup \mathcal{C}_{ord}(F^*, D^*)$ with minimum *average cost*, set $F^* \leftarrow F^* \cup \{i\}$ and $D^* \leftarrow D^* \cup A$, and assign all the clients $j \in A$ to the facility $i$. At this point, if $D^* = D$, then we terminate the algorithm. Otherwise, we proceed to the next round.

Intuitively, in each round the above greedy algorithm picks a cluster with minimum average *cost*, which is defined in such a way that takes into account the opening cost of a facility the first time some client gets assigned to it. Let $\mathcal{C}$ denote the collection of all clusters picked by this algorithm, across all the rounds. Note: (1) Every client $j \in A$ belongs to exactly one cluster in $\mathcal{C}$. (2) Let $\mathcal{C}(i) \subseteq \mathcal{C}$ denote the subset of clusters in $\mathcal{C}$ which contain a given facility $i \in F$. If $\mathcal{C}(i) \neq \emptyset$, then exactly one of the clusters in $\mathcal{C}(i)$ is critical. We refer to a collection $\mathcal{C}$ which satisfy these two properties as a *clustering*. It is easy to check that a clustering $\mathcal{C}$ defines a valid solution to the concerned instance of the facility location problem, where the objective value of the solution is given by $obj(\mathcal{C}) := \sum_{C \in \mathcal{C}} cost(C)$.

Note that if the above algorithm picks an *satellite* cluster $C = (i, A)$ in a given round, then w.l.o.g. we can assume that $|A| = 1$. This is because if $|A| > 1$, then we can only decrease the average cost of $C$ by deleting from $A$ all but the one client that is closest to facility $i$. Accordingly, from now on we will only consider those satellite clusters that have exactly one client.

Our dynamic algorithm is based on a *relaxation* of the above static greedy algorithm, where in each round we have the flexibility of picking a cluster with *approximately* minimum average cost. The output of this relaxed greedy algorithm corresponds to what we call a *nice clustering* (see Section 2.2 for a formal definition). We now present a high-level, informal overview of our dynamic algorithm.

**Preprocessing:** Upon receiving an input $(F, D)$ at preprocessing, we run the relaxed greedy algorithm which returns a nice clustering $\mathcal{C}$. We assign each cluster $C \in \mathcal{C}$ to an (not necessarily positive) integer *level* $\ell(C) \in \mathbb{Z}$ such that $cost_{avg}(C) = \Theta(2^{\ell(C)})$. Thus, the level of a cluster encodes its average cost upto a constant multiplicative factor. Define the level of a client $j \in C$ to be $\ell(j) := \ell(C)$.

**Handling an update:** When a client $j$ gets deleted from $D$, we simply delete the concerned client from the cluster in $\mathcal{C}$ it appears in. Similarly, when a client $j$ gets inserted into $D$, we arbitrarily assign it to any open facility $i$ by creating an satellite cluster $(C = (i, \{j\})$ at the minimum possible level $k \geq \Theta(\log d_{ij})$. At this point, we check if the clustering $\mathcal{C}$ being maintained by our algorithm still remains *nice*, i.e., whether $\mathcal{C}$ can correspond to an output of the relaxed greedy algorithm on the current input. If the answer is yes, then our dynamic algorithm is done with handling the current update. Thus, for the rest of this discussion, assume that the clustering $\mathcal{C}$ is no longer *nice*.

It turns out that in this event we can always identify a cluster $C = (i, A)$ (not necessarily part of $\mathcal{C}$) and a level $k \in \mathbb{Z}$ such that: $cost_{avg}(C) \leq O(2^k)$ and $\ell(j) > k$ for all clients $j \in A$. We refer to such a cluster $C = (i, A)$ as a *blocking cluster*. Intuitively, the existence of a blocking cluster is a certificate that the current clustering $\mathcal{C}$ is not nice. This is because the relaxed greedy algorithm constructs the levels in a bottom-up manner, in increasing order of average costs of the clusters that get added to the solution in successive rounds. Accordingly, the relaxed greedy algorithm would have added the cluster $C$ to its solution at level $\leq k$ before proceeding to level $k + 1$, and this contradicts the fact that a client $j \in A$ appears at level $> k$ in the current clustering $\mathcal{C}$.

As long as there is a blocking cluster $C = (i, A)$ at some level $k$, we update the current clustering $\mathcal{C}$ by calling a subroutine FIX-BLOCKING$(C, k)$ which works as follows: (1) it adds the cluster $C$ to level $k$, and (2) it removes each client $j \in A$ from the cluster in $\mathcal{C}$ it belonged to prior to this step.

Next, note that because of step (2) above, some cluster $C' = (i', A')$ might lose one or more clients $j$ as they move from $C'$ to the newly formed cluster $C$, and this might increase the average cost of the cluster $C'$. Thus, we might potentially end up in a situation where a cluster $C' = (i', A')$ has $cost_{avg}(C') \gg 2^{\ell(C')}$. As long as this happens to be the case, we call a subroutine FIX-LEVEL$(C)$ whose job is to increase the level of the affected cluster $C'$ by one.

Our algorithm repeatedly calls the subroutines FIX-BLOCKING$(.,.)$ and FIX-LEVEL$(.)$ until there is no blocking cluster and every cluster $C \in \mathcal{C}$ has $cost_{avg}(C) = \Theta(2^{\ell(C)})$. At this point, we are guaranteed that the current clustering is nice and we are done with processing the concerned update.

The approximation ratio of our algorithm follows from the observation that a nice clustering corresponds to the output of the relaxed greedy algorithm in the static setting, which in turn gives a $O(1)$-approximation. We bound the amortized recourse by a careful token-based argument.

## 2.2 Nice clustering

In Section 2.1, we explained that a *nice clustering* corresponds to an output of the relaxed greedy algorithm in the static setting. We now give a formal definition of this concept by introducing four invariants. Specifically, we define a clustering $\mathcal{C}$ to be nice iff it satisfies Invariants 1, 2, 3 and 4.

Every cluster $C \in \mathcal{C}$ is assigned a (not necessarily positive) integer *level* $\ell(C) \in \mathbb{Z}$. For any cluster $C = (i, A) \in \mathcal{C}$ and any client $j \in A$, we define the level of the client $j$ to be $\ell(j) := \ell(C)$.

**Invariant 1.** *For every cluster $C \in \mathcal{C}$, we have $cost_{avg}(C) < 2^{\ell(C)}$.*

The relaxed greedy algorithm constructs these levels in a "bottom-up" manner: it assigns the relevant clusters to a level $k$ before moving on to level $k + 1$. This holds because the algorithm picks the clusters in (approximately) increasing order of their average costs. Also, before picking an satellite cluster $C = (i, \{j\})$, the algorithm ensures that facility $i$ is open. This leads to the invariant below.

**Invariant 2.** *Consider any facility $i \in F$ with $\mathcal{C}(i) \neq \emptyset$. Then the unique critical cluster $C^* \in \mathcal{C}(i)$ satisfies: $\ell(C^*) \leq \ell(C)$ for all $C \in \mathcal{C}(i)$.*

The next invariant captures the fact that in an output of the relaxed greedy algorithm, if a client $j \in D$ gets assigned to a facility $i \in F$ then $\ell(j) \geq \Theta(\log d_{ij})$.

**Invariant 3.** *For every cluster $C = (i, A) \in \mathcal{C}$ and every client $j \in A$, we have $\ell(j) \geq \kappa_{ij}^*$, where $\kappa_{ij}^*$ is the unique level s.t. $2^{\kappa_{ij}^* - 4} \leq d_{ij} < 2^{\kappa_{ij}^* - 3}$.*

Finally, we formulate an invariant which captures the fact that the relaxed greedy algorithm picks the clusters in an (approximately) increasing order of their average costs. Towards this end, we define the notion of a *blocking cluster*, as described below.

**Definition 1.** *Consider any clustering $\mathcal{C}$, any level $k \in \mathbb{Z}$, and any cluster $C = (i, A)$ that does* not *necessarily belong to $\mathcal{C}$. The cluster $C$ is a* blocking cluster *at level $k$ w.r.t. $\mathcal{C}$ iff three conditions hold. (a) We have $cost_{avg}(C) < 2^{k-3}$. (b) For every $j \in A$, we have $\ell(j) > k \geq \kappa_{ij}^*$, where $\kappa_{ij}^*$ is the unique level s.t. $2^{\kappa_{ij}^* - 4} \leq d_{ij} < 2^{\kappa_{ij}^* - 3}$. (c) If $C$ is an satellite cluster, then there is a critical cluster $C^* \in \mathcal{C}(i)$ with $\ell(C^*) \leq k$. Else if $C$ is a critical cluster, then $k \leq \ell(C')$ for all $C' \in \mathcal{C}(i)$.*

Intuitively, if $C = (i, A)$ is a blocking cluster at level $k$ w.r.t. $\mathcal{C}$, then the relaxed greedy algorithm should have formed the cluster $C$ at some level $k' < k$ before proceeding to construct the subsequent levels which currently contain all the clients in $A$. This leads us to the invariant below.[3]

**Invariant 4.** *There is no blocking cluster w.r.t. the clustering $\mathcal{C}$ at any level $k \in \mathbb{Z}$.*

**Remark.** Consider a cluster $C = (i, A) \in \mathcal{C}$ at level $\ell(C) = k$. If $cost_{avg}(C) \ll 2^k$, then it is not difficult to show that there exists a subset of clients $A' \subseteq A$ such that $C'' = (i, A')$ is a blocking cluster at level $k - 1$. This, along with Invariant 1, implies that $cost_{avg}(C) = \Theta(2^{\ell(C)})$ for all $C \in \mathcal{C}$.

The next theorem holds since a nice clustering corresponds to the output of the relaxed greedy algorithm, and since the (original) greedy algorithm is known to have an approximation ratio of $O(1)$. We defer the proof of Theorem 2 to Appendix B.

**Theorem 2.** *Any nice clustering forms a $O(1)$-approximate optimal solution to the concerned input instance of the facility location problem.*

### 2.3 Description of Our Dynamic Algorithm

In the dynamic setting, we simply maintain a nice clustering. To be more specific, w.l.o.g. we assume that $D = \emptyset$ at preprocessing. Subsequently, during each update, a client gets inserted into / deleted from the set $D$. We now describe how our algorithm handles an update.

**Handling the deletion of a client** $j$: Suppose that client $j$ belonged to the cluster $C = (i, A) \in \mathcal{C}$ just before getting deleted. We set $A \leftarrow A \setminus \{j\}$, and then we call the subroutine described in Figure 1.

**Handling the insertion of a client** $j$: We identify any open facility $i \in F$.[4] Let $\kappa_{ij}^* \in \mathbb{Z}$ be the unique level such that $2^{\kappa_{ij}^* - 4} \leq d_{ij} < 2^{\kappa_{ij}^* - 3}$. Let $C = (i, A)$ be the unique critical cluster with facility $i$. If $\kappa_{ij}^* \leq \ell(C)$, then we set $A \leftarrow A \cup \{j\}$. In this case the client $j$ becomes part of the critical cluster $C$ at level $\ell(C)$. Else if $\kappa_{ij}^* > \ell(C)$, then we create a new satellite cluster $(i, \{j\})$ at level $\ell((i, \{j\})) = \kappa_{ij}^*$, and set $\mathcal{C} \leftarrow \mathcal{C} \cup \{(i, j)\}$. Next, we call the subroutine described in Figure 1.

**The subroutine** FIX-CLUSTERING(.): The insertion / deletion of a client might lead to a violation of Invariant 4 or Invariant 1 (the procedure described in the preceding two paragraphs ensure that Invariant 2 and Invariant 3 continue to remain satisfied). The subroutine FIX-CLUSTERING(.) handles this issue. It repeatedly checks whether Invariant 4 or Invariant 1 is violated, and accordingly calls FIX-BLOCKING$(C, k)$ or FIX-LEVEL$(C)$. Each of these last two subroutines has the property that it does not lead to any new violation of Invariant 2 or Invariant 3. Thus, when the WHILE loop in Figure 1 terminates, all the invariants are restored and we end up with a nice clustering.

---

| |
|---|
| 1. WHILE either Invariant 4 or Invariant 1 is violated: |
| 2.    IF Invariant 4 is violated, THEN |
| 3.        Find a cluster $C = (i, A)$ that is blocking at level $k$ (say). |
| 4.        Call the subroutine FIX-BLOCKING$(C, k)$. |
| 5.    ELSE IF Invariant 1 is violated, THEN |
| 6.        Find a cluster $C = (i, A)$ (say) that violates Invariant 1. |
| 7.        Call the subroutine FIX-LEVEL$(C)$. |

Figure 1: FIX-CLUSTERING(.).

**The subroutine** FIX-BLOCKING$(C, k)$: This subroutine works as follows. We first add the cluster $C = (i, A)$ to the current clustering. Towards this end, we set $\ell(C) \leftarrow k, \mathcal{C} \leftarrow \mathcal{C} \cup \{C\}$, and perform the following operations for all clients $j \in A$.

- Let $C_j' = (i_j', A_j')$ be the cluster the client $j$ belonged to just before the concerned call to the subroutine FIX-BLOCKING$(C, k)$. Set $A_j' \leftarrow A_j' \setminus \{j\}$.

---

[3]The condition $\ell(j) \geq \kappa_{ij}^*$ guarantees that $C$, in some sense, is a minimal blocking cluster. This is because if we remove a client violating this condition from the cluster $C$, then $cost_{avg}(C)$ can never become $> 2^{k-\mu}$. We will use this condition while analyzing the amortized recourse of our dynamic algorithm.

[4]If $j$ is the first client being inserted, then there is no existing open facility. In this case, we identify the facility $i \in F$ which minimizes $d_{ij} + f_i$, create a critical cluster $C = (i, \{j\})$ and assign it to the level $k \in \mathbb{Z}$ such that $d_{ij} + f_i \in [2^{k-4}, 2^{k-3})$.

At this point, we fork into one of two cases. *Case (a):* If $C$ is an satellite cluster, then there is nothing further that needs to be done, and we terminate the call to this subroutine. *Case (b):* If $C = (i, A)$ is a critical cluster, then let $C' = (i, A')$ denote the unique critical cluster with facility $i$ just before the concerned call to the subroutine. Since there has to be exactly one critical cluster with facility $i$, we now convert the cluster $C' = (i, A')$ into (possibly) multiple satellite clusters by performing the following operations for all clients $j \in A'$.

- Create an satellite cluster $(i, \{j\})$ at level $\ell((i, \{j\})) = \ell(C')$, and set $\mathcal{C} \leftarrow \mathcal{C} \cup \{(i, \{j\})\}$.

Finally, we remove the previous critical cluster $C'$ from the current clustering, by setting $\mathcal{C} \leftarrow \mathcal{C} \setminus \{C'\}$, and then terminate the call to this subroutine.

**The subroutine** FIX-LEVEL($C$)**:** Suppose that the cluster $C = (i, A)$ is currently at level $k = \ell(C)$. We first deal with the corner case where $\mathcal{C}(i) = \{C\}$ and $A = \emptyset$. In this case, we simply set $\mathcal{C} \leftarrow \mathcal{C} \setminus \{C\}$, and then terminate the call to this subroutine. Next, we check if $C$ is a critical cluster. If the answer is yes, then the cluster $C$ *absorbs* all the existing satellite clusters involving facility $i$ at level $k$. To be more specific, for all satellite clusters $(i, \{j\})$ at level $k$, we set $A \leftarrow A \cup \{j\}$ and $\mathcal{C} \leftarrow \mathcal{C} \setminus \{(i, \{j\})\}$. This ensures that Invariant 2 continues to remain satisfied. Finally, at this point if it continues to be the case that $cost_{avg}(C) \geq 2^k$, then we move the cluster $C$ up to level $k + 1$, by setting $\ell(C) \leftarrow \ell(C) + 1$. We now terminate the call to this subroutine.

## 2.4 Analysis of Our Dynamic Algorithm

The approximation guarantee of our dynamic algorithm follows from Theorem 2 and Theorem 3. We defer the proof of Theorem 3 to Appendix A.

**Theorem 3.** *The clustering $\mathcal{C}$ maintained by our dynamic algorithm always remains nice.*

For the rest of this section, we focus on bounding the amortized recourse of our algorithm. Towards this end, whenever a client moves up (resp. down) one level, we say that the algorithm performs one unit of *up-work* (resp. *down-work*).

**Lemma 1.** *During any sequence of $t$ updates, the total recourse of the algorithm is at most $t$ plus $O(1)$ times the total number of units of down-work that get performed.*

*Proof.* First, note that the total facility-recourse is upper bounded, within a constant multiplicative factor, by the total client-recourse. This is because if we open a new facility, then it necessarily implies that at least one client gets reassigned to the newly open facility.

Next, observe that a client $j$ gets assigned to a new facility $i$ only if one of the following two events occurs. (1) The client $j$ gets inserted during an update. (2) The client $j$ participates in a blocking cluster $C = (i, A)$ at level $k$ (say), and the algorithm calls the subroutine FIX-BLOCKING($C, k$). The total recourse incurred due to events of type (1) is at most $t$. On the other hand, during an event of type (2) the concerned client $j$ moves down from its current level $\ell(j)$ to a smaller level $k$, as per condition (b) in Definition 1. Thus, the total recourse incurred due to events of type (2) is at most the total units of down-work performed by the algorithm. The lemma follows. $\qquad \square$

It now remains to bound the total units of down-work performed by our algorithm. We achieve this by means of a *token based* argument, which in turn, requires us to first introduce the notion of an *epoch* of a cluster, and then bound the total units of *up-work* performed by our algorithm during an epoch.

**Epoch:** Observe that when a concerned cluster $C = (i, A)$ first becomes part of the clustering $\mathcal{C}$ at some level (say) $k$, we have $cost_{avg}(C) < 2^{k-3}$ (see condition (a) in Definition 1). Subsequently, during its entire lifetime, the level of the cluster $C$ can only increase (this can happen because of calls to the subroutine FIX-LEVEL($C$)). Accordingly, we partition the lifetime of the concerned cluster $C = (i, A)$ into multiple *epochs*, where an epoch refers to a maximal time-interval during which the level of the cluster does *not* increase. To be a bit more specific, suppose that the lifetime of cluster $C$ consists of epochs $\{0, 1, \ldots, \lambda\}$, for some integer $\lambda \geq 0$. An epoch $r \in [0, \lambda]$ begins when the cluster $C = (i, A)$ arrives at level $k + r$, and ends when the cluster moves up from level $k + r$ to level $k + r + 1$. Let $A_r$ denote the state of the set $A$ just before the end of epoch $r$. The algorithm performs $|A_r|$ units of up-work for the cluster $C = (i, A)$ in epoch $r$. We will now upper bound the total units of up-work performed for the cluster $C$ during its entire lifetime, which is given by $\sum_{r=0}^{\lambda} |A_r|$. The proof of Lemma 2 appears in Appendix C.

**Lemma 2.** *Consider a cluster $C = (i, A)$ that first became part of the clustering $\mathcal{C}$ at level $k$. During its entire lifetime, our algorithm performs at most $f_i/2^{k-2}$ units of up-work on this cluster $C$.*

**Tokens:** Recall that the opening costs of the facilities and the distances between the clients and facilities are all polynomially bounded by $m = |F|$. Hence, there are at most $O(\log m)$ levels in the clustering $\mathcal{C}$. To be more specific, there are two integral parameters $L, U \in \mathbb{Z}$, with $0 \leq U - L = O(\log m)$, such that $\ell(j) \in [L, U]$ for all clients $j \in D$ at all times. To analyse the amortised recourse of our algorithm, we now associate some *tokens* with each client in the following manner.

$$\text{For each client } j \in D, \text{ there is a token at each level } k \in [L, \ell(j)]. \tag{1}$$

Observation 1 holds since there are only $O(\log m)$ levels in the clustering $\mathcal{C}$.

**Observation 1.** *Whenever a client is inserted, it leads to the* creation *of at most $O(\log m)$ tokens. In contrast, whenever a client is deleted, it leads to the* release *of at most $O(\log m)$ tokens. We assume that these released tokens get* deposited *into a special* bank-account.

**Observation 2.** *Whenever the algorithm performs one unit of down-work, it* releases *one token. We assume that $1/2$ of this released token gets* deposited *into the special* bank-account, *and the remaining $1/2$ of the token* disappears *after paying for the cost of the down-work being performed. In contrast, whenever the algorithm performs one unit of up-work, it* withdraws *one token from the special bank-account to ensure that (1) continues to hold.*

We assume that just before preprocessing, the special bank-account had zero balance. We will show that throughout the duration of our algorithm, the balance in this bank-account always remains non-negative. Towards this end, observe that the algorithm withdraws tokens from this account only when it performs up-work, and it performs up-work only when some cluster increases its level by one. Accordingly, we focus on a cluster $C = (i, A)$ that first becomes part of the clustering $\mathcal{C}$ at time $\tau_0$ (say). Suppose that $C$ was assigned to level $k$ (say) at time $\tau_0$. Let $A^*$ denote the state of $A$ at time $\tau_0$. As per condition (a) of Definition 1, at time $\tau_0$ we have:

$$\frac{f_i}{|A^*|} \leq cost_{avg}(C) = \frac{f_i + \sum_{j \in A^*} d_{ij}}{|A^*|} < 2^{k-3}.$$

Hence, we get $|A^*| \geq f_i/2^{k-3}$. The next observation holds since every client in $A^*$ had to move down at least one level before the cluster $C$ is formed at level $k$, as per condition (b) of Definition 1.

**Observation 3.** *Just before a cluster $C$ becomes part of the clustering $\mathcal{C}$ for the first time (say, at a level $k$), the algorithm deposits at least $f_i/2^{k-4}$ many tokens into the special bank-account.*

Lemma 2 implies that at most $f_i/2^{k-2}$ tokens get withdrawn from the bank-account because of the cluster $C$. By Observation 3, this is at most the number of tokens that get deposited into the same account just before the cluster $C$ is formed at time $\tau_0$. It follows that the bank-account never runs into negative balance. Theorem 4 now follows from Observations 1, 2 and Lemma 1.

**Theorem 4.** *Starting from an empty set of clients, our algorithms spends at most $O(t \log m)$ total recourse to handle a sequence of $t$ updates. So, the amortised recourse of our algorithm is $O(\log m)$.*

## 2.5 Implementation of our algorithm in $\widetilde{O}(m)$ amortized update time.

In Appendix D we show how to implement our algorithm in amortized $\widetilde{O}(m)$ time per update. Here, we briefly describe how to quickly (i.e., in $\widetilde{O}(m)$ time) check whether Invariant 4 is satisfied.

For every facility $i$ and level $k$, let $S(i, k)$ denote the sequence of clients that are at a level larger than $k$, in increasing order of their distances from the facility $i$. The key observation is this: If there exists a blocking cluster involving facility $i$ at level $k$, then there must exist a blocking cluster (with the same facility and at the same level) whose clients form a prefix of $S(i, k)$. This has the following implication. Suppose that we explicitly maintain these sequences $\{S(i, k)\}$. Then given a specific pair $(i, k)$, we can determine in only $\widetilde{O}(1)$ time (via a simple binary search and some standard data structures) whether or not there is a blocking cluster involving facility $i$ at level $k$. Finally, note that we can indeed maintain the sequences $\{S(i, k)\}$ by paying a factor $\widetilde{O}(m)$ overhead in our update time. This is because whenever a client changes its level (or gets inserted / deleted) we need to modify at most $mL = \widetilde{O}(m)$ of these sequences $\{S(i, k)\}$, where $L$ is the number of levels.

In Appendix D, we present a more fine tuned data structure, where we discretize the distances in powers of $(1 + \epsilon)$. However, the main idea behind the data structure is the same as described above.

# 3 Experimental Evaluation

**Datasets** We experiment with three classic datasets [5] from UCI library Dua and Graff [2017]: KDD-Cup Stolfo et al. [2000] ($311,029$ points of dimension 74) and song Bertin-Mahieux et al. [2011] ($515,345$ points of dimension 90) Census Kohavi et al. [1996] ($2,458,285$ points of dimension 68). We did not alter in any way the dimensions of the data.

For each dataset, we only keep a $5,000$ points for each experiment as suffices to show the merits and limitations of the different approaches; we also confirm that the relative behavior of the algorithms remains the same over the whole sequence of updates on one of the dataset that we consider. We consider the $L_2$ distances of the embedding, and add $1/5000$ to each distance to avoid having distances zero while not disturbing the structure of the instance.

**Order of updates.** We consider insertions and deletions of points under the sliding window model, where we order the points according to some permutation, and then we consider an interval (a.k.a., window) of $\ell = 1,000$ indices, which we slide from the beginning to the end of the ordered points and at each step form an instance containing the points that lie within the window limits; that is, as we slide the window, the new point that gets within the limits of the window is treated as an insertion, while each point that falls outside the window limits is treated as a deletion.

**Choice of facilities.** We consider two different processes for choosing the set of facilities and their opening cost. The first way is to pick uniformly at random $5\%$ of the point in the instance, remove them from the set of clients, and make them facilities. In the second variant we use $50\%$ of the points as facilities. Using these two processes we generate six families of instances: Census-5%, Census-50%, KDD-Cup-5%, KDD-Cup-50%, song-5%, and song-50%. These instances aim to cover the settings where the set of facilities is very restrictive or very broad, in order to assess the behavior of the algorithms in both of these settings.

We set the facility opening costs within a range to ensure that the problem does not become too easy; that is, we don't want to the trivially good solutions of either opening only a single, or opening all facilities. We set the facility weights as follows: for a given instance, we calculated the median distance of clients to their nearest facility, and multiplied this number by $100$.

**Algorithms** We compare our algorithm against two main competing algorithms: 1) Following each point insertion or deletion, we rerun from scratch the offline Greedy algorithm Jain et al. [2003], which we call GREEDYOFF and 2) The state of the art $O(\log(m))$ approximation algorithm for the fully-dynamic facility location problem from Guo et al. [2020], which we call GKLX (from the authors last names). We also compare our algorithm to the baseline that simply maintains open the nearest facility of each of client. We call this baseline NEARESTFACILITY.

The behavior of our algorithm depends on two parameters: $\mu$ and $\epsilon$. The parameter $\epsilon$ defines the base of the exponential bucketing scheme, and the parameter $\mu$ defines the level $\kappa_{ij}^*$ (see Invariant 3), so that $(1+\epsilon)^{\kappa_{ij}^*-\mu-1} \leq d_{ij} < (1+\epsilon)^{\kappa_{ij}^*-\mu}$. In Section 2.2 we set $\epsilon = 1$ and $\mu = 3$ for ease of analysis. Both of these control parameters trade-off between running time, approximation, and recourse. While our theoretical part covers the case where $\mu \geq 3$ and $\epsilon = 1$, we go beyond what is proven and explore setting values $\mu \in \{1,3\}$ and $\epsilon \in \{0.05, 1\}$. We use NICECLUSTERING($\mu = 1, \epsilon = 0.05$), NICECLUSTERING($\mu = 1, \epsilon = 1$), NICECLUSTERING($\mu = 3, \epsilon = 0.05$), NICECLUSTERING($\mu = 3, \epsilon = 1$) to refer to the different configurations of our algorithm. We note that we did not try to optimize $\mu$ and $\epsilon$ to minimize the approximation guarantee (the main goal was to get any constant) for simplicity of the presentation and the analysis, although we believe this is doable.

We defer the interested reader to Appendix D for implementation details.

**Setup.** All of our code is written in C++ and is available online [6]. We used a e2-standard-16 Google Cloud instance, with 16 cores, 2.20GHz Intel(R) Xeon(R) processor, and 64 GiB main memory.

---

[5] None of these datasets is ideal for our setting as they lack information that determines the order of insertions and deletions. However, we are not aware of any targeted datasets that are openly available. We note that our datasets and setup are typical in studies of dynamic algorithms (see, e.g., Cohen-Addad et al. [2019]).

[6] `https://github.com/google-research/google-research/tree/master/fully_dynamic_facility_location`

### 3.1 Results

Throughout this section, as the relative behavior of the algorithms is similar among the different datasets, we use the KDD-Cup-5% and KDD-Cup-50% datasets to analyze the performance of the algorithm, and summarize the behavior of the algorithms for the rest of the datasets (we defer the detailed description and the plots of the remaining datasets to Appendix E).

Given that our process of creating the instances contain randomized decisions, we repeat each of our experiments 10 times and report the mean relative behavior of the algorithms in Appendix F. For visualization purposes, we discuss the outcome of a single experiment; repeating the experiment multiple times does not alter significantly the relative behavior of the algorithms.

Moreover, we assessed the performance of NICECLUSTERING($\mu = 1, \epsilon = 0.05$), it performs identically to NICECLUSTERING($\mu = 3, \epsilon = 0.05$) in terms of solution cost and time, but exhibits slightly larger recourse. Hence, we omit it to avoid complicating the plots and tables. Finally, we only run GREEDYOFF on the datasets KDD-Cup-5% and KDD-Cup-50%, and only once, for illustration purposes as it is very computationally expensive to run.

**Solution cost.** We analyze the different algorithms in terms of the cost of their solution. Figure 2 summarizes the results for both settings where we consider $5\%$ and $50\%$ of the points to be facilities.

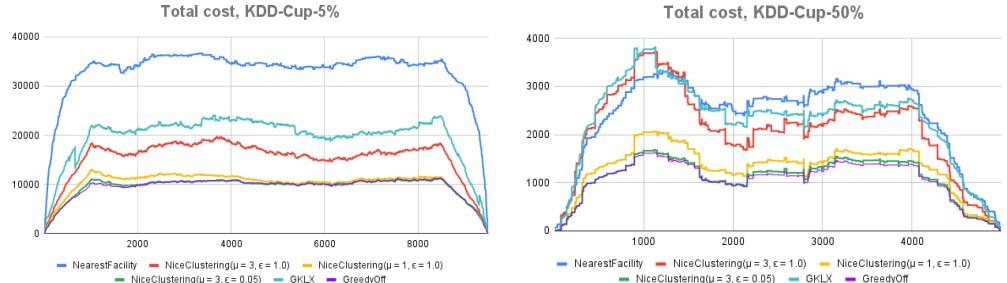

Figure 2: The solution cost by the different algorithms over the whole sequence of updates, on KDD-Cup-5% and KDD-Cup-50%.

As expected, the smaller the values of $\mu$ and $\epsilon$ in NICECLUSTERING, the better the solution quality. NICECLUSTERING($\mu = 3, \epsilon = 0.05$) perform almost identically to GREEDYOFF, which is an algorithm that is very computationally expensive. Over 10 repetitions, NICECLUSTERING($\mu = 3, \epsilon = 0.05$) performs roughly $17\% - 29\%$ better and $55\% - 67\%$ better than GKLX on KDD-Cup-5% and KDD-Cup-50%, respectively. The $> 55\%$ happens because in this particular dataset there are a few facilities with small distances to many clients, GKLX prioritizes minimizing opening costs rather than connection cost in this case. On the other datasets NICECLUSTERING($\mu = 3, \epsilon = 0.05$) outperforms GKLX by $14\% - 30\%$ (see Appendix E).

**Recourse.** Next, we study the recourse occurred by each of the algorithms that we consider on the datasets KDD-Cup-5% and KDD-Cup-50%. Figure 3 summarizes the results of this experiment.

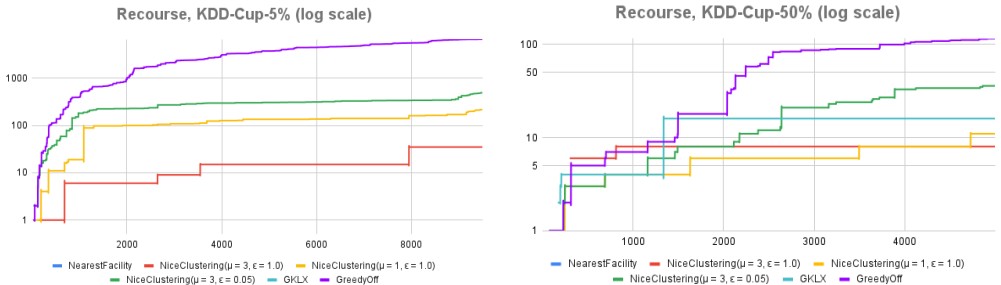

Figure 3: The cumulative recourse incurred by the algorithms that we consider over the whole sequence of updates, on KDD-Cup-5% (left), and KDD-Cup-50% (right). Missing lines imply 0 recourse throughout. The plot is in log scale.

While all algorithms, except GREEDYOFF, have limited recourse (considering the window size is $1,000$ and the length of the update sequence is $9,500$ and $5,000$ for KDD-Cup-5% and KDD-

Cup-50%, respectively), GKLX performs the best (NEARESTFACILITY trivially has no recourse). Naturally, smaller values of $\mu$ and $\epsilon$ in our algorithm make perform closer to GREEDYOFF (which has no recourse guarantees) in terms of cost, implying higher recourse. While GKLX performs better than NICECLUSTERING on KDD-Cup-5%, they are comparable on KDD-Cup-50%. Repeating the experiment 10 times show similar relative behavior.

**Running time.** As expected, our algorithm is slower compared to GKLX. NICECLUSTERING($\mu = 3, \epsilon = 1$) NICECLUSTERING($\mu = 1, \epsilon = 1$) are one order of magnitude slower. This is dues to the fact that GKLX, apart from the initial preprocessing phase and determining the nearest facility to each client, spends only polylogarithmic time per update as it operates on a tree structure of logarithmic depth. In fact, after the preprocessing phase, GKLX does not need to store the connections between clients and facilities; it suffices to remember the nearest facility to each client. The reduced space allows better cashing and fewer main memory accesses. On the other hand, our algorithm stores explicitly the connection costs of clients and visit them regularly. NICECLUSTERING($\mu = 3, \epsilon = 0.05$) is slower by another order of magnitude, due to the increased number of level in our algorithm. We refer to Appendix E.1 for the running time plots. Finally, GREEDYOFF is over two orders of magnitude slower compared to the slowest of the variants of our algorithm.

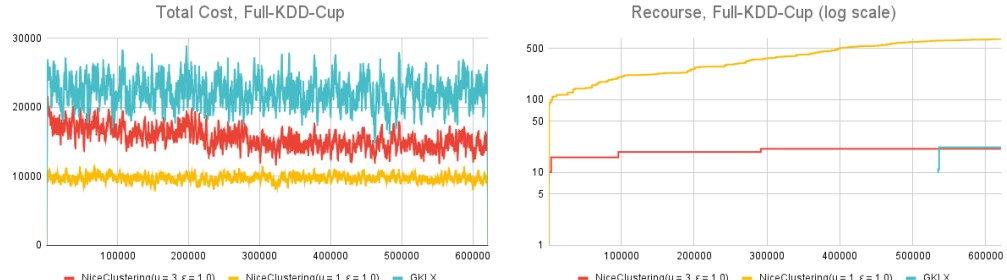

Figure 4: The solution cost by NICECLUSTERING($\mu = 3, \epsilon = 1$), NICECLUSTERING($\mu = 1, \epsilon = 1$), GKLX over the whole update sequence, on Full-KDD-Cup (left), and the total recourse incurred by these algorithms, in log scale, on Full-KDD-Cup (right). Missing lines imply no recourse throughout.

**Consistency over the long sequence of updates** To showcase that the relative behavior of the algorithms is not affected by our choice to restrict each datasets to the first 5000 points, we run the algorithm on whole KDD-Cup dataset consisting of 311,029 points, where we choose 250 points as facilities (the same absolute number as in the case of KDD-Cup-5%). Similarly to the rest of the experiments, we consider insertions and deletions of points under the sliding window model (with $\ell = 1000$). We refer to this instance as Full-KDD-Cup. We compare the most competitive algorithms, that is NICECLUSTERING($\mu = 3, \epsilon = 1$), NICECLUSTERING($\mu = 1, \epsilon = 1$), and GKLX in terms of the total cost, and recourse; the results are shown in Figure 4. The conclusions from the previous experiments on the restricted dataset hold also in this experiment. However, we observe two behaviors. First, the performance of GKLX fluctuates more compared to NICECLUSTERING($\mu = 3, \epsilon = 1$) and NICECLUSTERING($\mu = 1, \epsilon = 1$). This can be attributed to the fact that the algorithm doesn't adapt rapidly to the changes in the instance. To a lesser extend, the behavior of NICECLUSTERING($\mu = 3, \epsilon = 1$) also fluctuates more than NICECLUSTERING($\mu = 1, \epsilon = 1$), for the same reasons. The second observation is that while NICECLUSTERING($\mu = 3, \epsilon = 1$) and GKLX incur very little recourse, while NICECLUSTERING($\mu = 1, \epsilon = 1$) continuously incurs (limited) recourse. These two observations are related. This implies that NICECLUSTERING($\mu = 1, \epsilon = 1$) adapts more rapidly to the changing dataset so that it is able to maintain a better solution.

**Conclusion of Experiments.** We showed that our algorithm outperforms, often significantly, in terms of solution cost the GKLX algorithm, but in some cases exhibits an increase in recourse (while still being a small absolute number) and it is also slower. We believe the our algorithm is a good fit for applications where solution quality is very important, and one wants to maintain a stable solution in reasonable time. Given that our algorithm performs almost in par with GREEDYOFF, it becomes a good choice in cost-sensitive applications on dynamic data.

## Acknowledgments and Disclosure of Funding

Sayan Bhattacharya is supported by Engineering and Physical Sciences Research Council, UK (EPSRC) Grant EP/S03353X/1.

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

### On theoretical results

Q (a): Did you state the full set of assumptions of all theoretical results?

A: yes.

Q (b): Did you include complete proofs of all theoretical results?

A: yes.

**On experiments**

Q (a): Did you include the code, data, and instructions needed to reproduce the main experimental results (either in the supplemental material or as a URL)?

A: No. We will the code is made available.

Q (b): Did you specify all the training details (e.g., data splits, hyperparameters, how they were chosen)?

A: There is no training in this paper.

Q (c): Did you report error bars (e.g., with respect to the random seed after running experiments multiple times)?

A: Yes, we repeated the experiments 10 times, and report the mean and standard deviation in the Appendix F.

Q (d): Did you include the amount of compute and the type of resources used (e.g., type of GPUs, internal cluster, or cloud provider)?

A: Yes. The experimental section reports the type of machine that we used.

**On using existing assets.**

Q (a): If your work uses existing assets, did you cite the creators?

A: Yes. We cited the datasets that we used.

Q (b): Did you mention the license of the assets?

A: No. These datasets have no individual licences in the UCI repository.

Q (c): Did you include any new assets either in the supplemental material or as a URL?

A: No. We released the code.

Q (d): Did you discuss whether and how consent was obtained from people whose data you're using/curating?

A: Yes. They the data are available at the UCI machine learning repository, and no additional licence is reported.

Q (e): Did you discuss whether the data you are using/curating contains personally identifiable information or offensive content?

A: No. This is clear since the data are simply embeddings.

