# A Proof of Theorem 3

By induction hypothesis, suppose that the clustering $\mathcal{C}$ satisfies Invariant 1, Invariant 2, Invariant 4 and Invariant 3 just before a given update. While handling the update, our dynamic algorithm calls the subroutine in Figure 1. The main WHILE loop in Figure 1 immediately ensures that Invariant 1 and Invariant 4 continue to hold when our dynamic algorithm is done with processing the update.

Moving on to Invariant 2, it is easy to verify that this invariant is not violated by our dynamic algorithm at any point in time. Specifically, this invariant does not get violated by any step in Figure 1, or by the steps performed immediately before calling the subroutine described in Figure 1.

It now remains to prove that our algorithm always satisfy Invariant 3. Towards this end, fix any client $j \in D$. If the client $j$ gets assigned to facility $i \in F$ immediately after getting inserted, then the manner in which our algorithm handles an insertion ensures that $\ell(j) \geq \kappa_{ij}^*$ at that point in time. Subsequently, consider each of the following two three of events.

(1) The client $j$ remains assigned to the facility $i$, but moves down to a smaller level (say) $k$. This happens only if there is a blocking cluster $C = (i, A)$ with $j \in A$ at level $k$, and we call the subroutine FIX-BLOCKING$(C, k)$. Hence, Definition 1 ensures that $k \geq \kappa_{ij}^*$. Thus, we continue to have $\ell(j) = k \geq \kappa_{ij}^*$ immediately after this event.

(2) The client $j$ remains assigned to the facility $i$, but moves up to a larger level. Clearly, this event cannot lead to a violation of Invariant 3.

(3) The client $j$ gets reassigned to a different facility $i'$, by joining a cluster at level $k$ (say) that has facility $i'$ as its center. This happens only if the client $j$ is part of a blocking cluster $C' = (i', A')$ at level $k$, with $j' \in A'$, and we call the subroutine FIX-BLOCKING$(C', k)$. Hence, Definition 1 ensures that $k \geq \kappa_{i'j}^*$. Thus, we continue to have $\ell(j) = k \geq \kappa_{i'j}^*$ immediately after this event.

The above discussion implies that Invariant 3 continues to remain satisfied all the time, even when the algorithm is handling an update. This concludes the proof of Theorem 3.

# B Proof of Theorem 2

We first recall the LP relaxation for the facility location problem:

$$\text{Minimize} \sum_i f_i \cdot y_i + \sum_j \sum_i d_{ij} \cdot x_{ij} \tag{2}$$

$$\text{s.t.} \sum_{i \in F} x_{ij} \geq 1 \qquad \text{for all } j \in D \tag{3}$$

$$x_{ij} \leq y_i \qquad \text{for all } i \in F, j \in D \tag{4}$$

$$y_i, x_{ij} \geq 0 \qquad \text{for all } i \in F, j \in D \tag{5}$$

The dual LP is given below.

$$\text{Maximize} \sum_{j \in D} \alpha_j \tag{6}$$

$$\text{s.t.} \sum_{j \in D} \beta_{ij} \leq f_i \qquad \text{for all } i \in F \tag{7}$$

$$\alpha_j - \beta_{ij} \leq d_{ij} \qquad \text{for all } i \in F, j \in D \tag{8}$$

$$\alpha_j, \beta_{ij} \geq 0 \qquad \text{for all } i \in F, j \in D \tag{9}$$

We will prove Theorem 2 using the primal-dual method. Specifically, we will show that given any nice clustering, we can come up with a pair of feasible primal and dual solutions whose objectives are within a $O(1)$ multiplicative factor of each other. For the rest of the proof, fix any nice clustering $\mathcal{C}$.

Based on the clustering $\mathcal{C}$, we construct the following natural primal solution $(x, y)$. For all $i \in F$, we set $y_i = 1$ if facility $i$ is open (i.e., if $\mathcal{C}(i) \neq \emptyset$); otherwise, we set $y_i = 0$. Similarly, for all

$i \in F, j \in D$, we set $x_{ij} = 1$ if the clustering $\mathcal{C}$ assigns client $j$ to facility $i$; otherwise, we set $x_{ij} = 0$. It is easy to check that this gives us a feasible primal solution. Next, based on the clustering $\mathcal{C}$, we construct an (infeasible) dual solution $(\alpha, \beta)$ as described in Figure 5.

```
1.    FOR ALL clusters C = (i, A) ∈ C:
2.        FOR ALL clients j ∈ A:
3.            Set α_j ← 2^ℓ(C).
4.            FOR ALL facilities i' ∈ F:
5.                Set β_ij ← max (0, α_j/2^10 − d_ij).
```

Figure 5: Constructing an (infeasible) dual solution $(\alpha, \beta)$ based on a nice clustering $\mathcal{C}$.

We now show that the primal objective at $(x, y)$ is at most the dual objective at $(\alpha, \beta)$.

**Lemma 3.** *We have $\sum_{i \in F} f_i \cdot x_i + \sum_{i \in F, j \in D} d_{ij} \cdot y_{ij} \leq \sum_{j \in D} \alpha_j$.*

*Proof.* The cost of the primal solution $(x, y)$ is equal to the total cost of all the clusters in the dual solution $(\alpha, \beta)$. Thus, we have:

$$\sum_{i \in F} f_i \cdot x_i + \sum_{i \in F, j \in D} d_{ij} \cdot y_{ij} = \sum_{C \in \mathcal{C}} cost(C). \tag{10}$$

Next, fix any cluster $C = (i, A) \in \mathcal{C}$, and observe that:

$$cost(C) = |C| \cdot cost_{avg}(C) \leq |C| \cdot 2^{\ell(C)} = \sum_{j \in A} \alpha_j. \tag{11}$$

The lemma now follows from (10) and (11) and the observation that each client $j \in A$ appears in exactly one cluster $C \in \mathcal{C}$. □

The next lemma shows that $(\alpha/2^{10}, \beta)$ is a feasible dual solution.

**Lemma 4.** *For all $j \in D$, define $\hat{\alpha}_j := \alpha_j/2^{10}$. Then $(\hat{\alpha}, \beta)$ is a feasible solution to LP (6).*

Theorem 2 follows from Lemma 3 and Lemma 4. The proof of Lemma 4 appears in Appendix B.1.

### B.1   Proof of Lemma 4

Step 5 of Figure 5 ensures that the solution $(\hat{\alpha}, \beta)$ satisfies constraint (8) of the dual LP. It now remains to show that the solution $(\hat{\alpha}, \beta)$ also satisfies constraint (7) of the dual LP. This is achieved by Lemma 6. In other words, Lemma 4 follows from Lemma 6.

**Lemma 5.** *Consider any facility $i \in F$ and clients $j, j' \in D$ with $\alpha_j \geq d_{ij}$ and $\alpha_{j'} \geq d_{ij'}$. Then we have: $\alpha_j/2^4 \leq d_{ij} + 2\alpha_{j'}$.*

*Proof.* If $\alpha_j \leq \alpha_{j'}$, then the lemma trivially holds; so we assume that $\alpha_j > \alpha_{j'}$ throughout the rest of the proof. From step 3 of Figure 5, it follows that each client $j^* \in D$ appears at a level $\ell(j^*) = \log(\alpha_{j^*})$ in the clustering $\mathcal{C}$. Accordingly, since $\alpha_{j'} < \alpha_j$, we have $\ell(j') \leq \ell(j) - 1$.

Let $i'$ denote the facility to which the client $j'$ gets assigned in the nice clustering $\mathcal{C}$. Now we claim that $d_{i'j} \geq \alpha_j/2^4$. For the sake of contradiction, suppose $d_{i'j} < \alpha_j/2^4$. We will show that $C := (i', \{j\})$ forms an satellite blocking cluster at level $k := \ell(j) - 1$. Towards this end, we note:

- (a) $cost_{avg}(C) = d_{i'j} < \alpha_j \cdot 2^{-4} = 2^{\ell(j)-4} = 2^{k-3}$.

- (b) $\ell(j) > k \geq \kappa^*_{i'j}$, where $\kappa^*_{i'j}$ is the unique level such that $2^{\kappa^*_{i'j}-4} \leq d_{i'j} < 2^{\kappa^*_{i'j}-3}$. Here, the first inequality holds because $k := \ell(j) - 1$, whereas the second inequality holds because $2^{\kappa^*_{i'j}-4} \leq d_{i'j} < 2^{k-3}$ as per condition (a) above.

- (c) Since $i'$ is the center of a cluster at level $\ell(j')$ and $\mathcal{C}$ is a nice clustering, there exists a critical cluster $C' \in \mathcal{C}(i')$ with $\ell(C') \leq \ell(j') \leq \ell(j) - 1 = k$.

Conditions (a), (b) and (c) imply that $C = (i', \{j\})$ is an satellite blocking cluster at level $\ell(j) - 1$, as per Definition 1. This contradicts the fact that $\mathcal{C}$ is a nice clustering. Hence, we get:

$$d_{i'j} \geq \alpha_j/2^4. \tag{12}$$

Next, since client $j' \in D$ is assigned to facility $i' \in F$ in the nice clustering $\mathcal{C}$, by Invariant 3 we have $\kappa^*_{i'j'} \leq \ell(j')$. This implies that:

$$d_{i'j'} < 2^{\kappa^*_{i'j'}} \leq 2^{\ell(j')} = \alpha_{j'}. \tag{13}$$

Applying triangle inequality, we now obtain $d_{i'j} \leq d_{i'j'} + d_{ij'} + d_{ij} \leq d_{i'j'} + \alpha_{j'} + d_{ij} \leq 2 \cdot \alpha'_j + d_{ij}$ (the last inequality follows from (13)), and $d_{ij'} \leq \alpha_{j'}$ by our starting assumption). Thus, from (12) we infer that: $\alpha_j/2^4 \leq d_{i'j} \leq 2\alpha'_j + d_{ij}$. $\qquad\square$

**Lemma 6.** *For every facility $i \in F$, we have $\sum_{j \in D} \max\left(0, \frac{\alpha_j}{2^{10}} - d_{ij}\right) \leq f_i$.*

*Proof.* Fix a facility $i \in F$ for the rest of the proof. Define $C_i := \{j \in D \mid \frac{\alpha_j}{2^{10}} \geq d_{ij}\}$. Note that:

$$\sum_{j \in D} \max\left(0, \frac{\alpha_j}{2^{10}} - d_{ij}\right) = \sum_{j \in C_i} \left(\frac{\alpha_j}{2^{10}} - d_{ij}\right) \tag{14}$$

Accordingly, for the rest of the proof, we focus on upper bounding the right hand side of (14).

For ease of notations, we assume that $C_i = \{j_1, \ldots, j_q\}$ and w.l.o.g. $\alpha_{j_1} \leq \alpha_{j_2} \leq \cdots \leq \alpha_{j_q}$. Say that a client $j \in C_i$ is *big* if $d_{ij} \geq \alpha_{j_1}/2^4$ and *small* otherwise. Let $B \subseteq C_i$ and $S = C_i \setminus B$ respectively denote the set of all big and small clients in $C_i$. Observe that:

$$\alpha_{j_1} - d_{ij} \leq 2^4 \cdot d_{ij} \text{ for all } j \in B \implies \sum_{j \in B} (\alpha_{j_1} - d_{ij}) \leq 2^4 \cdot \sum_{j \in B} d_{ij}. \tag{15}$$

Next, we claim that:

$$\sum_{j \in S} (\alpha_{j_1} - d_{ij}) \leq 2^4 \cdot f_i + (2^4 - 1) \cdot \sum_{j \in S} d_{ij}. \tag{16}$$

For the sake of contradiction, suppose that (16) does not hold. Then we have:

$$|S| \cdot \alpha_{j_1} > 2^4 \left(f_i + \sum_{j \in S} d_{ij}\right) \implies \frac{\alpha_{j_1}}{2^4} > \frac{f_i + \sum_{j \in S} d_{ij}}{|S|}. \tag{17}$$

We now consider two possible cases.

*Case 1: There is no critical cluster in $\mathcal{C}(i)$ at level $\leq k := \ell(j_1) - 1 = \log(\alpha_{j_1}) - 1$.*

In this case, consider the critical cluster $C := (i, S)$. We will show that $C$ is a blocking cluster at level $k$ w.r.t. the clustering $\mathcal{C}$. To see why this is true, observe that:

- (a) $cost_{avg}(C) := \frac{f_i + \sum_{j \in S} d_{ij}}{|S|} < \frac{\alpha_{j_1}}{2^4} = \frac{2^{\ell(j_1)}}{2^4} = 2^{k-3}$.

- (b) Consider any client $j \in S$. Since $\alpha_j \geq \alpha_{j_1}$, we have $\ell(j) = \log(\alpha_j) \geq \log(\alpha_{j_1}) = \ell(j_1) > k$. Furthermore, since $j \in S$, we have $d_{ij} < \alpha_{j_1}/2^4 = 2^{k-3}$. This implies that $k \geq \kappa^*_{ij}$, where $\kappa^*_{ij}$ is the unique level such that $2^{\kappa^*_{ij}-4} \leq d_{ij} < 2^{\kappa^*_{ij}-3}$. To summarize, we infer that $\ell(j) > k \geq \kappa^*_{ij}$ for all clients $j \in S$.

- (c) We have $\ell(C') > k$ for all clusters $C' \in \mathcal{C}(i)$. This follows from Invariant 2 and our assumption that there is no critical cluster in $\mathcal{C}(i)$ at level $\leq k$.

Conditions (a), (b) and (c) above imply that $C$ is a critical blocking cluster at level $k$ w.r.t. the clustering $\mathcal{C}$, as per Definition 1. But this contradicts the fact that $\mathcal{C}$ is a nice clustering. Thus, we conclude that (16) holds in this case.

*Case 2: There is a critical cluster $C^* \in \mathcal{C}(i)$ at level $\ell(C^*) \leq k$.*

In this case, let $j \in S$ be a client such that $d_{ij} = \min_{j' \in S} d_{ij'}$. Consider the satellite cluster $C := (i, \{j\})$. We will show that $C$ is a blocking cluster at level $k := \ell(j_1) - 1$ w.r.t. the clustering $\mathcal{C}$. To see why this is true, observe that:

- (a) $cost_{avg}(C) := d_{ij} < \frac{\alpha_{j_1}}{2^4} = \frac{2^{\ell(j_1)}}{2^4} = 2^{k-3}$.

- (b) Since $\alpha_j \geq \alpha_{j_1}$, we have $\ell(j) = \log(\alpha_j) \geq \log(\alpha_{j_1}) = \ell(j_1) > k$. Furthermore, since $j \in S$, we have $d_{ij} < \alpha_{j_1}/2^4 = 2^{k-3}$. This implies that $k \geq \kappa^*_{ij}$, where $\kappa^*_{ij}$ is the unique level such that $2^{\kappa^*_{ij}-4} \leq d_{ij} < 2^{\kappa^*_{ij}-3}$. To summarize, we infer that $\ell(j) > k \geq \kappa^*_{ij}$.

- (c) There is a critical cluster $C^* \in \mathcal{C}(i)$ at level $\ell(C^*) \leq k$.

Conditions (a), (b) and (c) above imply that $C$ is an satellite blocking cluster at level $k$ w.r.t. the clustering $\mathcal{C}$, as per Definition 1. But this contradicts the fact that $\mathcal{C}$ is a nice clustering. Thus, we conclude that (16) holds in this case as well.

To summarize, we have proved that (16) holds. Thus, from (15) and (16), we get:

$$
\begin{aligned}
\sum_{j \in C_i} (\alpha_{j_1} - d_{ij}) &= \sum_{j \in B} (\alpha_{j_1} - d_{ij}) + \sum_{j \in S} (\alpha_{j_1} - d_{ij}) \\
&\leq 2^4 \cdot \sum_{j \in B} d_{ij} + 2^4 \cdot f_i + (2^4 - 1) \cdot \sum_{j \in S} d_{ij} \\
&\leq 2^4 \cdot f_i + 2^4 \cdot \sum_{j \in C_i} d_{ij}.
\end{aligned}
\tag{18}
$$

Now consider any client $j \in C_i$. Since $j, j_1 \in C_i$, by definition we have $d_{ij} \leq \alpha_j$ and $d_{ij_1} \leq \alpha_{j_1}$. Accordingly, Lemma 5 gives us:

$$
\alpha_j/2^4 \leq d_{ij} + 2 \cdot \alpha_{j_1} \implies \alpha_j \leq 2^4 \cdot d_{ij} + 2^5 \cdot \alpha_{j_1}.
\tag{19}
$$

Summing (19) over all the clients $j \in C_i$, we get:

$$
\begin{aligned}
\sum_{j \in C_i} \alpha_j &\leq \sum_{j \in C_i} \left(2^4 \cdot d_{ij} + 2^5 \cdot \alpha_{j_1}\right) = (2^4 + 2^5) \sum_{j \in C_i} d_{ij} + 2^5 \sum_{j \in C_i} (\alpha_{j_1} - d_{ij}) \\
&\leq (2^4 + 2^5) \sum_{j \in C_i} d_{ij} + 2^9 f_i + 2^9 \sum_{j \in C_i} d_{ij} \qquad \text{(follows from (18))} \\
&\leq 2^{10} \sum_{j \in C_i} d_{ij} + 2^{10} f_i \implies \sum_{j \in C_i} \left(\frac{\alpha_j}{2^{10}} - d_{ij}\right) \leq f_i.
\end{aligned}
\tag{20}
$$

The lemma now follows from (14) and (20). $\qquad \square$

## C   Proof of Lemma 2

We use the notations introduced in the paragraph just before the statement of Lemma 2. Note that during its entire lifetime, our algorithm performs $\sum_{r=0}^{\lambda} |A_r|$ units of up-work on the cluster $C$. Accordingly, Lemma 2 follows from Claim 1.

**Claim 1.** *We have $\sum_{r=0}^{\lambda} |A_r| \leq f_i/2^{k-2}$.*

*Proof.* Consider a given round $r \in [0, \lambda]$. For all $j \in A_r$, we have $d_{ij} < 2^{\kappa^*_{ij}-3} \leq 2^{k+r-3}$, where the last inequality follows from Invariant 3 (this is because every client $j \in A_r$ is at level $k + r$ just before the end of epoch $r$). The cluster $C = (i, A)$ moves up to level $k + r + 1$ because $cost_{avg}(C) > 2^{k+r}$. Thus, we conclude that:

$$
\frac{f_i + 2^{k+r-3} \cdot |A_r|}{|A_r|} \geq \frac{f_i + \sum_{j \in A_r} d_{ij}}{|A_r|} = cost_{avg}((i, A_r)) > 2^{k+r}.
$$

Multiplying both sides of the above inequality by $|A_r|$ and then rearranging the terms, we get:

$$
f_i > \left(2^{k+r} - 2^{k+r-3}\right) \cdot |A_r| \geq 2^{k+r-1} \cdot |A_r|.
$$

So, we have $|A_r| \leq \frac{f_i}{2^{k+r-1}}$ for all $r \in [0, \lambda]$. This gives us: $\sum_{r=0}^{\lambda} |A_r| \leq \sum_{r=0}^{\lambda} \frac{2 f_i}{2^{k+r}} \leq \frac{f_i}{2^{k-2}}$. $\quad \square$

# D Implementation details

In this section we describe how to implement our algorithm to perform only $\tilde{O}(m)$ amortized time per update, and also the cases where we deviate from the straightforward implementation as described in Section 2.3.

**Client insertions.** Following the insertion of a client $j$, we first try to connect it to its nearest open facility $i$ (by either adding it to the critical cluster centered at $i$, or create the satellite cluster $(i, \{j\})$), instead trying to connect it to any open facility $i \in F$ as described in Section 2.3.

**Efficient implementation of Routine** FIX-CLUSTERING$(.)$. We describe how to implement the routine so that, it requires $\tilde{O}(m)$ amortized time per update. In particular, we describe an implementation that takes $\tilde{O}(m)$ per update, plus $O(m)$ time per each level change of a client. The latter is still withing the $\tilde{O}(m)$ amortized bound, as each client changes levels $O(\log m)$ times on average (see Section 2.4).

First we need an efficient mechanism to detect whether there is a blocking cluster violating invariant 4, or if there exists a cluster violating invariant 1. Then, we are going to describe how we implement the subroutines FIX-BLOCKING$(C, k)$ and FIX-LEVEL$(C)$ that restore the invariants of the algorithm. We describe our data structure as if we had a parameter $\epsilon$ that defines the base of the exponential bucketing scheme, despite setting it to $\epsilon \leftarrow 1$ in Section 2.3.

**Checking for violations of invariants 4 and 1.** Notice, that by simply maintaining the average cost of each critical cluster one can simply iterate over all critical clusters (which are $O(m)$) and check whether Invariant 1 is violated. Satellite clusters need to be checked only at their creation time, as their set of clients and subsequently their average cost does not change. Hence, we don't need to iterate over the satellite clusters at each iteration inside FIX-CLUSTERING$(.)$. In fact, each satellite cluster is checked iteratively until it is placed at the right level following its creation (either by inserting a new client, or by converting a critical cluster into an satellite one), and the time for all of these check is accounted in the bound of the up-work and down-work, which is $O(\log m)$ amortized per update.

Checking for violations of Invariant 4 efficiently requires an additional data structure. Conceptually, for each facility $i$, we maintain a two-dimensional matrix $\mathcal{W}$ of counters of dimensions $\log_{(1+\epsilon)}(\Delta) \times (\ell_{max} - \ell_{min})$, where $\ell_{max}, \ell_{min}$ the maximum and minimum level that are non-empty at any point in time $\Delta$ the ratio of the largest distance over the smallest distance between a client and a facility. The entry $\mathcal{W}[x, y]$ contains the number of clients that are at distance in the interval $[(1 + \epsilon)^{x-1}, (1 + \epsilon)^x)$ from facility $i$ and are currently placed at level $y$. This matrix has size at most $O(\log^2_{(1+\epsilon)}(\Delta))$ (which is $O(\log^2(m))$ as we assume $\Delta$ is polynomially bounded by $m$), but can be sparsified by maintaining a two dimensional hash table containing only the entries with positive counter values, while still maintaining constant time access with high probability. Although we implement the sparse version of this data structure, for ease of description we assume that we maintain the full matrix $\mathcal{W}$. Using the matrix $\mathcal{W}$ for facility $i$, we check whether there exists a blocking cluster at level $\ell$ centered at a facility $i$ as follows. We iterate over all entries of $\mathcal{W}[x, y]$ such that $y > \ell$, and in increasing order of the $x$ coordinate. Each time we visit an entry $\mathcal{W}[x, y]$ we check whether the following inequality holds:

$$\frac{f_i + \left( \sum_{x'=x_{min}}^{x} \sum_{y'=\ell+1}^{y_{max}} W[x', y'] \cdot (1 + \epsilon)^{y'} \right)}{\left( \sum_{x'=x_{min}}^{x} \sum_{y'=\ell+1}^{y_{max}} 1 \right)} \leq (1 + \epsilon)^{\ell},$$

and if it does then we determine that facility $i$ forms a blocking cluster at level $\ell$. By maintaining appropriate running summations, we can iterate over the appropriate entries of the matrix, each time testing the aforementioned inequality, in time linear in the size of the matrix. Notice that due to the bucketisation of the distances in powers of $(1 + \epsilon)$, we lose another factor of $(1 + \epsilon)$ in the approximation; that is we say that a cluster is blocking at level $\ell$ only if its average cost is below $(1 + \epsilon)^{(\ell-1)}$ instead of if its average cost is below $(1 + \epsilon)^{\ell}$, which increases the approximation by another $(1 + \epsilon)$ factor. Finally, notice that to maintain the matrix $\mathcal{W}$ of each facility updated, we only

need to spend constant time each time a client changes levels (as the distance between a facility and a client does not change). That is, for each client changing levels, we spend $O(m)$ time to update the matrices of all facilities. A client can change levels at most $O(\log m)$ times, as described in Section 2.4. Hence, one can check whether there exists a blocking cluster in amortized time $\tilde{O}(m)$ over the sequence of all updates. In order to identify the actual blocking cluster one can do that in time proportional to the size of the blocking cluster: One needs to maintain the neighbors of each facility in increasing distance at each level and visit them in increasing distance from the facility and only at the levels that are eligible (that is, levels greater then $\ell$).

**Implementing subroutines** Fix-Blocking$(C, k)$ **and** Fix-Level$(C)$**.** The subroutines Fix-Blocking$(C, k)$ and Fix-Level$(C)$ perform the necessary changes to the maintained clustering to restore any violations to the invariants of the algorithm. The only two scenarios that causes a client to be assigned to a different cluster at the same level are 1) the case where operation Fix-Level$(C)$ increases the level of a critical cluster, which causes the critical cluster to merge with other satellite clusters of the destination level, and 2) the case where a critical cluster gets converted into a set of satellite clusters at the same level. In both of these scenarios, we can charge the work to the time spent creating the satellite clusters by insertions of new (which is $O(1)$ per update), plus the time of creating the critical clusters which then get converted into satellite clusters and is at most $O(\log m)$ amortized per udpate. In all other cases, whenever a client changes the cluster to which it is assigned, it also changes levels. As discussed in Section 2.4, the number of changes of levels of clients is $O(\log m)$ amortized. This implies that the amortized time spent in the subroutines Fix-Blocking$(C, k)$ and Fix-Level$(C)$ is $O(\log m)$ per update.

# E    Missing experiments

## E.1    Missing experiments for **KDD-Cup** dataset

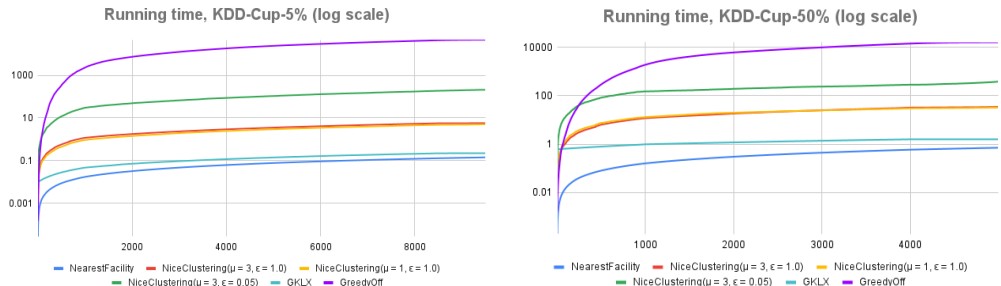

Figure 6: The running time of the algorithms that we consider, in seconds, on the datasets KDD-Cup-5% (left) and KDD-Cup-50% (right). The plot is in log scale.

## E.2    Experiments for **Census** dataset

In terms of cost, NICECLUSTERING$(\mu = 3, \epsilon = 0.05)$ algorithm performs overall the best on the Census-5% and Census-50% instances. In particular, it maintains solutions that are 20%-30% better compared to GKLX on average over the whole sequence of updates on the Census datasets. On the other hand NICECLUSTERING$(\mu = 1, \epsilon = 1)$ performs roughly 10% worse than NICECLUSTERING$(\mu = 3, \epsilon = 0.05)$, but better than GKLX. NICECLUSTERING$(\mu = 3, \epsilon = 1)$ runs roughly 70%-80% worse than NICECLUSTERING$(\mu = 3, \epsilon = 0.05)$, and has the worse performance in terms of cost overall. The behavior of the algorithms, for a single run, is summarized in Figure 7.

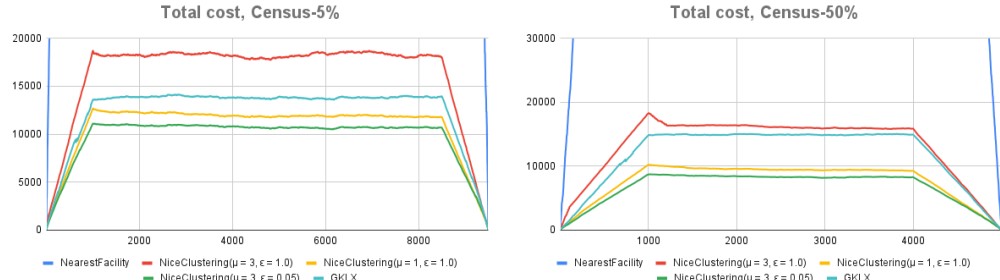

Figure 7: The total cost incurred by the algorithms that we consider, on the datasets Census-5% (left) and Census-50% (right). The full plot of the cost of NEARESTFACILITY is cut off to distinguish the performance of the rest of the algorithms, and is incomparable the rest of the algorithms.

In terms of recourse, NICECLUSTERING($\mu = 3, \epsilon = 0.05$) exhibits higher recourse than GKLX, by nearly two orders of magnitude on Census-5%, while GKLX11 incurs around $10\times$ recourse compared to GKLX. In all cases, recourse remains relatively low in absolute value (around $100$), especially considering the lengths of the update sequence and the instance sizes. The behavior of the algorithms, for a single run, is summarized in Figure 8.

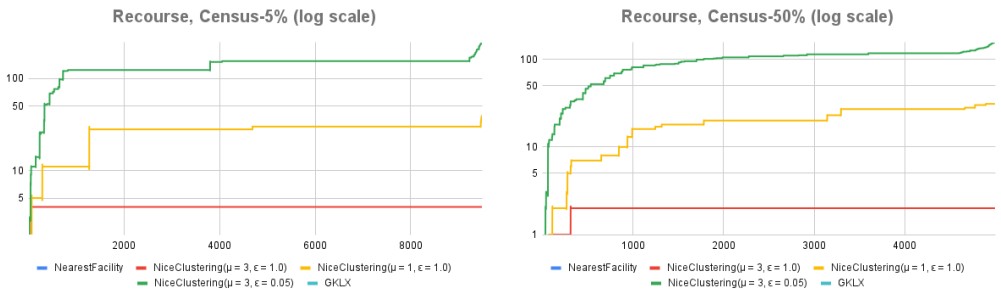

Figure 8: The cumulative recourse incurred by the algorithms that we consider, on the datasets Census-5% (left) and Census-50% (right). Missing lines imply 0 recourse throughout. The plot is in log scale.

The running time of the algorithms on the Census instances is very similar to that on the KDD-Cup instances, as described in the main body of the paper.

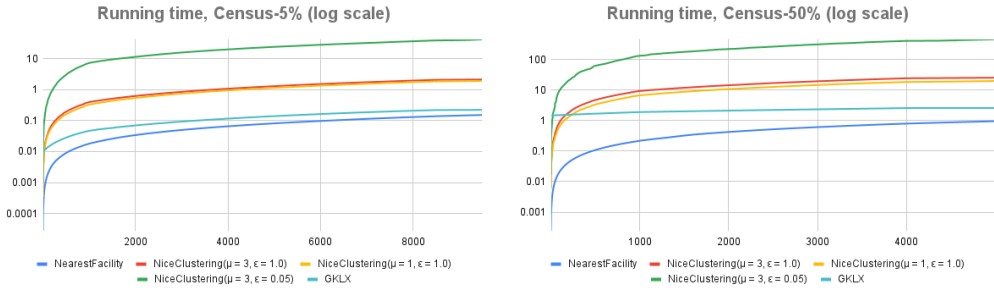

Figure 9: The running time of the algorithms that we consider, in seconds, on the datasets Census-5% (left) and Census-50% (right). The plot is in log scale.

## E.3 Experiments for **song** dataset

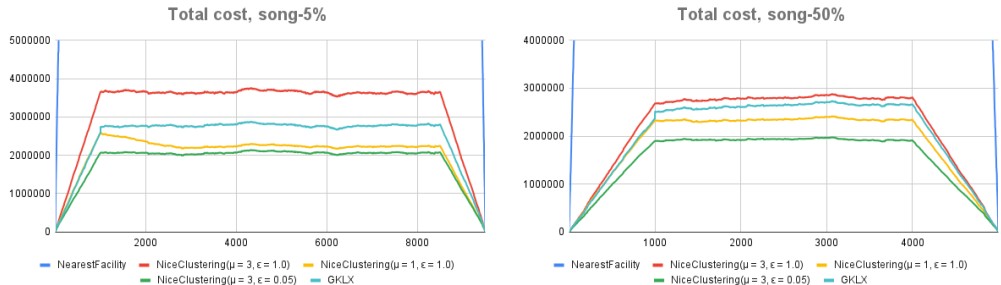

Figure 10: The total cost incurred by the algorithms that we consider, on the datasets **song**-5% (left) and **song**-50% (right). The full plot of the cost of NEARESTFACILITY is cut off to distinguish the performance of the rest of the algorithms, and is incomparable the rest of the algorithms.

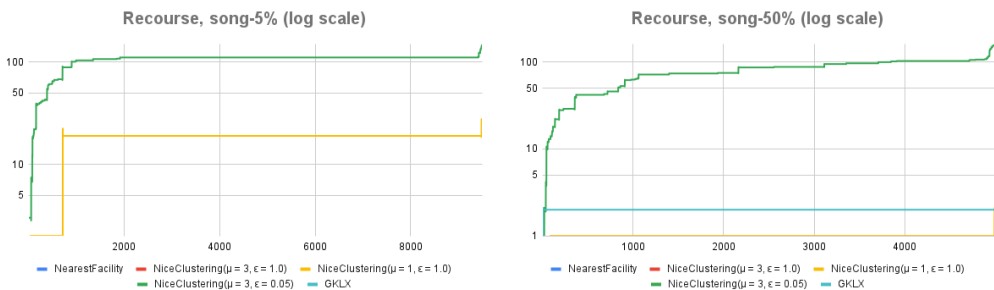

Figure 11: The cumulative recourse incurred by the algorithms that we consider, on the datasets **song**-5% (left) and **song**-50% (right). Missing lines imply 0 recourse throughout. The plot is in log scale.

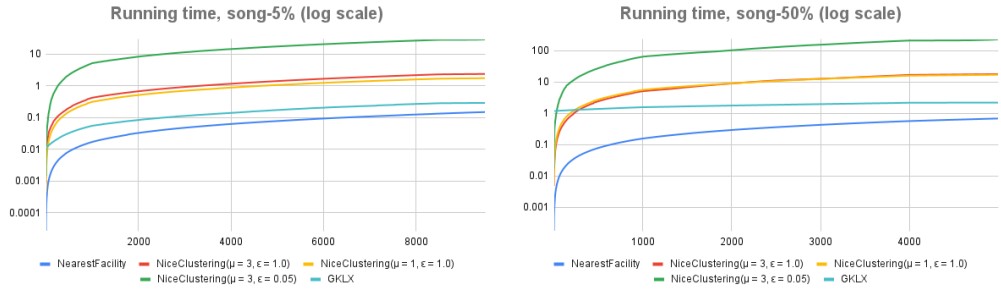

Figure 12: The running time of the algorithms that we consider, in seconds, on the datasets **song**-5% (left) and **song**-50% (right). The plot is in log scale.

## F    Comparing the relative behavior of the algorithms over multiple runs

**Comparison of algorithms**    In order to compare the relative behavior over multiple repetitions of each experiment that we consider, we proceed as follows. Consider the case where we want to compare the cost of the solution produced by algorithm $\mathcal{A}_1$ to the cost of the solution produced by algorithm $\mathcal{A}_2$. For each dataset that we consider and each repetition of an experiment we average the ratio $cost_i(\mathcal{A}_1)/cost_i(\mathcal{A}_2)$, for all $1 \leq i \leq max_u$, where $cost_i(\mathcal{A})$ the cost of the solution produced by algorithm $\mathcal{A}$ on the instance following the $i$-th update, and $max_u$ the total number

of updates. Denote this average by $\phi(\mathcal{A}_1, \mathcal{A}_2) = \left(\sum_{i=1}^{max_u} \frac{cost_i(\mathcal{A}_1)}{cost_i(\mathcal{A}_2)}\right)/max_u$. This gives us the relative behavior of the two algorithms with respect to the cost of the solution that they produce, on average, over the whole sequence of updates, for a single repetition.

We use the same methodology to compare the cumulative recourse of two algorithms, which we spell out for clarity. For each dataset that we consider and each repetition of an experiment we average the ratio $recourse_i(\mathcal{A}_1)/recourse_i(\mathcal{A}_2)$, for all $1 \leq i \leq max_u$, where $recourse_i(\mathcal{A})$ the cumulative recourse of the solution maintained by algorithm $\mathcal{A}$ following the $i$-th update. Denote this average by $\psi(\mathcal{A}_1, \mathcal{A}_2) = \left(\sum_{i=1}^{max_u} \frac{recourse_i(\mathcal{A}_1)+1}{recourse_i(\mathcal{A}_2)+1}\right)/max_u$. We add $+1$ to both the nominator and denominator to avoid division with 0.

To aggregate the behavior over multiple repetitions, we report the mean and the standard deviation of $\phi(\mathcal{A}_1, \mathcal{A}_2)$ and $\psi(\mathcal{A}_1, \mathcal{A}_2)$ over 10 repetitions of the experiment.

## F.1   Results

| | NF | NC$(3,1)$ | NC$(1,1)$ | NC$(3,0.05)$ | NC$(1,0.05)$ | GKLX |
|---|---|---|---|---|---|---|
| NF | 1 | 1.27 ($\pm$0.41) | 1.71 ($\pm$0.29) | 1.78 ($\pm$0.28) | 1.79 ($\pm$0.28) | 1.30 ($\pm$0.32) |
| NC$(3,1)$ | 1.01 ($\pm$0.00) | 1 | 1.50 ($\pm$0.28) | 1.58 ($\pm$0.29) | 1.58 ($\pm$0.29) | 1.10 ($\pm$0.21) |
| NC$(1,1)$ | 0.67 ($\pm$0.04) | 0.72 ($\pm$0.10) | 1 | 1.05 ($\pm$0.02) | 1.05 ($\pm$0.02) | 0.74 ($\pm$0.05) |
| NC$(3,0.05)$ | 0.64 ($\pm$0.04) | 0.69 ($\pm$0.10) | 0.96 ($\pm$0.01) | 1 | 1.00 ($\pm$0.00) | 0.71 ($\pm$0.06) |
| NC$(1,0.05)$ | 0.63 ($\pm$0.04) | 0.69 ($\pm$0.10) | 0.95 ($\pm$0.01) | 1.00 ($\pm$0.00) | 1 | 0.71 ($\pm$0.06) |
| GKLX | 0.92 ($\pm$0.09) | 0.98 ($\pm$0.13) | 1.39 ($\pm$0.12) | 1.45 ($\pm$0.15) | 1.46 ($\pm$0.16) | 1 |

Table 1: Relative behavior of every pair of algorithms that we consider, in terms of the cost of the solution that they produce. For a single entry in this table, let $\mathcal{A}_1$ (resp., $\mathcal{A}_2$ ) be the algorithm representing in the row (resp., column) of the entry. The entry reports the mean and standard deviation of $\phi(\mathcal{A}_1, \mathcal{A}_2)$, over 10 repetitions, for the dataset KDD-Cup-5%. $\pm$0.00 means the standard deviation is less than 0.005.

| | NF | NC$(3,1)$ | NC$(1,1)$ | NC$(3,0.05)$ | NC$(1,0.05)$ | GKLX |
|---|---|---|---|---|---|---|
| NF | 1 | 0.03 ($\pm$0.01) | 0.04 ($\pm$0.02) | 0.03 ($\pm$0.01) | 0.02 ($\pm$0.01) | 0.42 ($\pm$0.28) |
| NC$(3,1)$ | 78.70 ($\pm$32.30) | 1 | 2.61 ($\pm$0.78) | 1.42 ($\pm$0.78) | 1.36 ($\pm$0.84) | 30.20 ($\pm$30.70) |
| NC$(1,1)$ | 35.00 ($\pm$18.60) | 0.68 ($\pm$0.41) | 1 | 0.58 ($\pm$0.26) | 0.56 ($\pm$0.28) | 17.40 ($\pm$23.30) |
| NC$(3,0.05)$ | 183.00 ($\pm$196.00) | 2.93 ($\pm$3.29) | 5.80 ($\pm$0.00) | 1 | 1.02 ($\pm$0.27) | 82.10 ($\pm$157.00) |
| NC$(1,0.05)$ | 158.00 ($\pm$124.00) | 2.54 ($\pm$2.06) | 5.92 ($\pm$0.29) | 1.14 ($\pm$0.29) | 1 | 55.80 ($\pm$84.30) |
| GKLX | 9.06 ($\pm$5.35) | 0.14 ($\pm$0.07) | 0.34 ($\pm$0.12) | 0.17 ($\pm$0.12) | 0.15 ($\pm$0.11) | 1 |

Table 2: Relative behavior of every pair of algorithms that we consider, in terms of the cumulative recourse that they incur. For a single entry in this table, let $\mathcal{A}_1$ (resp., $\mathcal{A}_2$ ) be the algorithm representing in the row (resp., column) of the entry. The entry reports the mean and standard deviation of $\psi(\mathcal{A}_1, \mathcal{A}_2)$, over 10 repetitions, for the dataset KDD-Cup-5%. $\pm$0.00 means the standard deviation is less than 0.005.

|  | NF | NC(3,1) | NC(1,1) | NC(3,0.05) | NC(1,0.05) | GKLX |
|---|---|---|---|---|---|---|
| NF | 1 | 4.00 (±5.63) | 4.82 (±5.68) | 5.18 (±5.82) | 5.20 (±5.81) | 1.76 (±2.01) |
| NC(3,1) | 0.86 (±0.00) | 1 | 1.72 (±0.14) | 1.96 (±0.17) | 1.97 (±0.17) | 0.78 (±0.15) |
| NC(1,1) | 0.50 (±0.02) | 0.62 (±0.04) | 1 | 1.14 (±0.02) | 1.15 (±0.02) | 0.46 (±0.07) |
| NC(3,0.05) | 0.44 (±0.02) | 0.54 (±0.03) | 0.88 (±0.01) | 1 | 1.01 (±0.00) | 0.40 (±0.06) |
| NC(1,0.05) | 0.43 (±0.02) | 0.54 (±0.03) | 0.88 (±0.01) | 0.99 (±0.00) | 1 | 0.40 (±0.06) |
| GKLX | 1.19 (±0.27) | 2.18 (±2.43) | 3.13 (±2.66) | 3.47 (±2.71) | 3.49 (±2.71) | 1 |

Table 3: Relative behavior of every pair of algorithms that we consider, in terms of the cost of the solution that they produce. For a single entry in this table, let $\mathcal{A}_1$ (resp., $\mathcal{A}_2$) be the algorithm representing in the row (resp., column) of the entry. The entry reports the mean and standard deviation of $\phi(\mathcal{A}_1, \mathcal{A}_2)$, over 10 repetitions, for the dataset KDD-Cup-50%. ±0.00 means the standard deviation is less than 0.005.

|  | NF | NC(3,1) | NC(1,1) | NC(3,0.05) | NC(1,0.05) | GKLX |
|---|---|---|---|---|---|---|
| NF | 1 | 0.27 (±0.23) | 0.22 (±0.05) | 0.17 (±0.06) | 0.14 (±0.04) | 0.37 (±0.27) |
| NC(3,1) | 7.25 (±4.56) | 1 | 1.46 (±0.65) | 0.94 (±0.65) | 0.74 (±0.44) | 1.97 (±2.11) |
| NC(1,1) | 7.42 (±1.40) | 2.04 (±2.10) | 1 | 0.67 (±0.25) | 0.51 (±0.16) | 1.84 (±2.36) |
| NC(3,0.05) | 17.50 (±8.94) | 3.82 (±3.15) | 2.23 (±0.00) | 1 | 0.85 (±0.25) | 3.39 (±3.99) |
| NC(1,0.05) | 23.00 (±9.53) | 5.27 (±4.56) | 2.80 (±0.88) | 1.54 (±0.88) | 1 | 3.88 (±4.45) |
| GKLX | 21.80 (±15.20) | 4.55 (±5.74) | 2.71 (±1.12) | 1.66 (±1.12) | 1.21 (±1.01) | 1 |

Table 4: Relative behavior of every pair of algorithms that we consider, in terms of the cumulative recourse that they incur. For a single entry in this table, let $\mathcal{A}_1$ (resp., $\mathcal{A}_2$) be the algorithm representing in the row (resp., column) of the entry. The entry reports the mean and standard deviation of $\psi(\mathcal{A}_1, \mathcal{A}_2)$, over 10 repetitions, for the dataset KDD-Cup-50%. ±0.00 means the standard deviation is less than 0.005.

|  | NF | NC(3,1) | NC(1,1) | NC(3,0.05) | NC(1,0.05) | GKLX |
|---|---|---|---|---|---|---|
| NF | 1 | 4.80 (±0.72) | 7.24 (±0.40) | 7.91 (±0.38) | 8.00 (±0.38) | 6.15 (±1.30) |
| NC(3,1) | 0.23 (±0.00) | 1 | 1.57 (±0.17) | 1.73 (±0.20) | 1.75 (±0.19) | 1.33 (±0.25) |
| NC(1,1) | 0.15 (±0.00) | 0.66 (±0.05) | 1 | 1.10 (±0.02) | 1.11 (±0.02) | 0.85 (±0.13) |
| NC(3,0.05) | 0.13 (±0.00) | 0.60 (±0.05) | 0.91 (±0.02) | 1 | 1.01 (±0.00) | 0.77 (±0.11) |
| NC(1,0.05) | 0.13 (±0.00) | 0.59 (±0.05) | 0.90 (±0.01) | 0.99 (±0.00) | 1 | 0.76 (±0.11) |
| GKLX | 0.19 (±0.04) | 0.83 (±0.21) | 1.28 (±0.41) | 1.40 (±0.44) | 1.41 (±0.44) | 1 |

Table 5: Relative behavior of every pair of algorithms that we consider, in terms of the cost of the solution that they produce. For a single entry in this table, let $\mathcal{A}_1$ (resp., $\mathcal{A}_2$) be the algorithm representing in the row (resp., column) of the entry. The entry reports the mean and standard deviation of $\phi(\mathcal{A}_1, \mathcal{A}_2)$, over 10 repetitions, for the dataset Census-5%. ±0.00 means the standard deviation is less than 0.005.

|  | NF | NC(3,1) | NC(1,1) | NC(3,0.05) | NC(1,0.05) | GKLX |
|---|---|---|---|---|---|---|
| NF | 1 | 0.43 (±0.31) | 0.08 (±0.03) | 0.01 (±0.00) | 0.01 (±0.00) | 0.57 (±0.29) |
| NC(3,1) | 4.99 (±3.69) | 1 | 0.33 (±0.04) | 0.05 (±0.04) | 0.03 (±0.02) | 2.64 (±2.42) |
| NC(1,1) | 19.10 (±10.50) | 7.98 (±7.22) | 1 | 0.15 (±0.06) | 0.08 (±0.03) | 11.24 (±10.02) |
| NC(3,0.05) | 142.00 (±33.90) | 63.09 (±58.76) | 9.44 (±0.00) | 1 | 0.61 (±0.11) | 72.04 (±48.93) |
| NC(1,0.05) | 268.00 (±75.00) | 113.88 (±108.74) | 16.13 (±0.41) | 1.84 (±0.41) | 1 | 135.79 (±105.36) |
| GKLX | 9.95 (±7.62) | 3.48 (±3.88) | 0.64 (±0.04) | 0.07 (±0.04) | 0.04 (±0.03) | 1 |

Table 6: Relative behavior of every pair of algorithms that we consider, in terms of the cumulative recourse that they incur. For a single entry in this table, let $\mathcal{A}_1$ (resp., $\mathcal{A}_2$) be the algorithm representing in the row (resp., column) of the entry. The entry reports the mean and standard deviation of $\psi(\mathcal{A}_1, \mathcal{A}_2)$, over 10 repetitions, for the dataset Census-5%. ±0.00 means the standard deviation is less than 0.005.

|  | NF | NC(3,1) | NC(1,1) | NC(3,0.05) | NC(1,0.05) | GKLX |
|---|---|---|---|---|---|---|
| NF | 1 | 4.82 (±0.53) | 7.25 (±0.33) | 7.86 (±0.16) | 7.94 (±0.17) | 6.28 (±0.65) |
| NC(3,1) | 0.22 (±0.00) | 1 | 1.54 (±0.14) | 1.69 (±0.18) | 1.70 (±0.18) | 1.35 (±0.21) |
| NC(1,1) | 0.15 (±0.00) | 0.67 (±0.05) | 1 | 1.09 (±0.02) | 1.10 (±0.02) | 0.87 (±0.08) |
| NC(3,0.05) | 0.13 (±0.00) | 0.61 (±0.05) | 0.92 (±0.02) | 1 | 1.01 (±0.00) | 0.80 (±0.06) |
| NC(1,0.05) | 0.13 (±0.00) | 0.61 (±0.05) | 0.91 (±0.02) | 0.99 (±0.00) | 1 | 0.79 (±0.06) |
| GKLX | 0.17 (±0.01) | 0.79 (±0.11) | 1.18 (±0.14) | 1.28 (±0.13) | 1.30 (±0.13) | 1 |

Table 7: Relative behavior of every pair of algorithms that we consider, in terms of the cost of the solution that they produce. For a single entry in this table, let $\mathcal{A}_1$ (resp., $\mathcal{A}_2$) be the algorithm representing in the row (resp., column) of the entry. The entry reports the mean and standard deviation of $\phi(\mathcal{A}_1, \mathcal{A}_2)$, over 10 repetitions, for the dataset Census-50%. ±0.00 means the standard deviation is less than 0.005.

|  | NF | NC(3,1) | NC(1,1) | NC(3,0.05) | NC(1,0.05) | GKLX |
|---|---|---|---|---|---|---|
| NF | 1 | 0.30 (±0.28) | 0.07 (±0.03) | 0.01 (±0.01) | 0.01 (±0.00) | 0.29 (±0.29) |
| NC(3,1) | 8.28 (±6.88) | 1 | 0.42 (±0.10) | 0.09 (±0.10) | 0.07 (±0.08) | 2.43 (±4.62) |
| NC(1,1) | 21.35 (±7.78) | 5.10 (±4.33) | 1 | 0.19 (±0.12) | 0.14 (±0.10) | 5.19 (±7.03) |
| NC(3,0.05) | 167.01 (±102.33) | 62.30 (±95.60) | 9.33 (±0.00) | 1 | 0.69 (±0.12) | 33.70 (±29.64) |
| NC(1,0.05) | 274.23 (±141.82) | 95.60 (±132.00) | 15.59 (±0.42) | 1.65 (±0.42) | 1 | 44.07 (±36.83) |
| GKLX | 17.18 (±8.94) | 4.38 (±3.61) | 0.97 (±0.09) | 0.14 (±0.09) | 0.09 (±0.06) | 1 |

Table 8: Relative behavior of every pair of algorithms that we consider, in terms of the cumulative recourse that they incur. For a single entry in this table, let $\mathcal{A}_1$ (resp., $\mathcal{A}_2$) be the algorithm representing in the row (resp., column) of the entry. The entry reports the mean and standard deviation of $\psi(\mathcal{A}_1, \mathcal{A}_2)$, over 10 repetitions, for the dataset Census-50%. ±0.00 means the standard deviation is less than 0.005.

|  | NF | NC(3,1) | NC(1,1) | NC(3,0.05) | NC(1,0.05) | GKLX |
|---|---|---|---|---|---|---|
| NF | 1 | 6.95 (±1.14) | 8.70 (±0.60) | 9.77 (±0.53) | 9.99 (±0.45) | 7.84 (±0.77) |
| NC(3,1) | 0.16 (±0.00) | 1 | 1.30 (±0.20) | 1.46 (±0.18) | 1.51 (±0.21) | 1.17 (±0.19) |
| NC(1,1) | 0.13 (±0.00) | 0.79 (±0.08) | 1 | 1.14 (±0.03) | 1.17 (±0.03) | 0.90 (±0.04) |
| NC(3,0.05) | 0.11 (±0.00) | 0.70 (±0.05) | 0.88 (±0.02) | 1 | 1.03 (±0.01) | 0.80 (±0.04) |
| NC(1,0.05) | 0.11 (±0.00) | 0.68 (±0.06) | 0.86 (±0.02) | 0.98 (±0.01) | 1 | 0.78 (±0.04) |
| GKLX | 0.14 (±0.01) | 0.89 (±0.12) | 1.12 (±0.06) | 1.27 (±0.09) | 1.30 (±0.09) | 1 |

Table 9: Relative behavior of every pair of algorithms that we consider, in terms of the cost of the solution that they produce. For a single entry in this table, let $\mathcal{A}_1$ (resp., $\mathcal{A}_2$) be the algorithm representing in the row (resp., column) of the entry. The entry reports the mean and standard deviation of $\phi(\mathcal{A}_1, \mathcal{A}_2)$, over 10 repetitions, for the dataset song-5%. ±0.00 means the standard deviation is less than 0.005.

|  | NF | NC(3,1) | NC(1,1) | NC(3,0.05) | NC(1,0.05) | GKLX |
|---|---|---|---|---|---|---|
| NF | 1 | 0.92 (±0.23) | 0.30 (±0.35) | 0.02 (±0.01) | 0.01 (±0.00) | 0.32 (±0.23) |
| NC(3,1) | 2.00 (±3.00) | 1 | 0.39 (±0.02) | 0.03 (±0.02) | 0.02 (±0.02) | 0.40 (±0.31) |
| NC(1,1) | 7.26 (±3.65) | 6.34 (±3.52) | 1 | 0.10 (±0.05) | 0.06 (±0.03) | 1.68 (±0.77) |
| NC(3,0.05) | 118.00 (±49.10) | 106.11 (±54.66) | 35.90 (±0.00) | 1 | 0.66 (±0.26) | 30.45 (±29.23) |
| NC(1,0.05) | 222.00 (±100.00) | 210.56 (±114.13) | 55.97 (±1.22) | 2.23 (±1.22) | 1 | 44.09 (±31.32) |
| GKLX | 11.60 (±4.50) | 10.29 (±5.33) | 2.81 (±0.07) | 0.12 (±0.07) | 0.07 (±0.04) | 1 |

Table 10: Relative behavior of every pair of algorithms that we consider, in terms of the cumulative recourse that they incur. For a single entry in this table, let $\mathcal{A}_1$ (resp., $\mathcal{A}_2$) be the algorithm representing in the row (resp., column) of the entry. The entry reports the mean and standard deviation of $\psi(\mathcal{A}_1, \mathcal{A}_2)$, over 10 repetitions, for the dataset song-5%. ±0.00 means the standard deviation is less than 0.005.

| | NF | NC(3, 1) | NC(1, 1) | NC(3, 0.05) | NC(1, 0.05) | GKLX |
|---|---|---|---|---|---|---|
| NF | 1 | 6.35 (±1.35) | 8.33 (±0.52) | 9.70 (±0.42) | 9.89 (±0.44) | 7.96 (±0.59) |
| NC(3, 1) | 0.18 (±0.00) | 1 | 1.38 (±0.26) | 1.64 (±0.37) | 1.67 (±0.37) | 1.32 (±0.26) |
| NC(1, 1) | 0.13 (±0.00) | 0.76 (±0.09) | 1 | 1.18 (±0.03) | 1.20 (±0.03) | 0.96 (±0.03) |
| NC(3, 0.05) | 0.11 (±0.00) | 0.65 (±0.08) | 0.85 (±0.02) | 1 | 1.02 (±0.01) | 0.81 (±0.03) |
| NC(1, 0.05) | 0.11 (±0.00) | 0.63 (±0.08) | 0.84 (±0.02) | 0.98 (±0.01) | 1 | 0.80 (±0.03) |
| GKLX | 0.14 (±0.00) | 0.79 (±0.10) | 1.05 (±0.03) | 1.24 (±0.05) | 1.26 (±0.05) | 1 |

Table 11: Relative behavior of every pair of algorithms that we consider, in terms of the cost of the solution that they produce. For a single entry in this table, let $\mathcal{A}_1$ (resp., $\mathcal{A}_2$) be the algorithm representing in the row (resp., column) of the entry. The entry reports the mean and standard deviation of $\phi(\mathcal{A}_1, \mathcal{A}_2)$, over 10 repetitions, for the dataset song-50%. ±0.00 means the standard deviation is less than 0.005.

| | NF | NC(3, 1) | NC(1, 1) | NC(3, 0.05) | NC(1, 0.05) | GKLX |
|---|---|---|---|---|---|---|
| NF | 1 | 0.99 (±0.04) | 0.41 (±0.34) | 0.02 (±0.01) | 0.01 (±0.00) | 0.43 (±0.19) |
| NC(3, 1) | 1.12 (±0.36) | 1 | 0.42 (±0.01) | 0.02 (±0.01) | 0.01 (±0.00) | 0.43 (±0.19) |
| NC(1, 1) | 5.38 (±3.83) | 5.19 (±3.79) | 1 | 0.06 (±0.05) | 0.03 (±0.02) | 2.43 (±2.95) |
| NC(3, 0.05) | 134.83 (±82.51) | 132.00 (±81.40) | 46.79 (±0.00) | 1 | 0.59 (±0.17) | 63.67 (±97.23) |
| NC(1, 0.05) | 239.92 (±108.17) | 232.00 (±98.70) | 83.56 (±0.62) | 2.05 (±0.62) | 1 | 99.59 (±132.35) |
| GKLX | 9.20 (±4.21) | 8.84 (±3.74) | 3.12 (±0.06) | 0.10 (±0.06) | 0.05 (±0.02) | 1 |

Table 12: Relative behavior of every pair of algorithms that we consider, in terms of the cumulative recourse that they incur. For a single entry in this table, let $\mathcal{A}_1$ (resp., $\mathcal{A}_2$) be the algorithm representing in the row (resp., column) of the entry. The entry reports the mean and standard deviation of $\psi(\mathcal{A}_1, \mathcal{A}_2)$, over 10 repetitions, for the dataset song-50%. ±0.00 means the standard deviation is less than 0.005.