# OpenReview forum: "Efficient and Stable Fully Dynamic Facility Location"
_NeurIPS.cc/2022/Conference — NeurIPS 2022 Accept_

### Official Review · Reviewer_d29J · 2022-06-29

**Rating:** 7
**Confidence:** 3
**Soundness:** 3 good
**Presentation:** 3 good
**Contribution:** 3 good

**Summary:**

The authors develop the first fully dynamic algorithm for facility location with constant approximation guarantee and polylogarithmic recourse per update. This guarantees that the current solution does not change ``too much'' after any update. Prior to this this work, the best known fully dynamic algorithm achieved a logarithmic approximation guarantee and polylog update time. Their theoretical results are complemented with an experimental evaluation on real-world data.

**Questions:**

The authors answered convincingly my question about the polylog amortized running time.

**Limitations:**

I could not find any potential negative societal impact.

**Strengths And Weaknesses:**

+ the first constant approximation algorithm for facility location with polylog recourse per update
+ facility location is relevant to the machine learning community (it can be seen as the Lagrangian relaxation of k-median), while there have been increasing efforts in recent years to develop fully dynamic algorithms that are stable
+ the theoretical results are non-trivial and interesting
+ the paper is well written and relatively easy to follow

- the constant approximation ratio is relatively large (2^10, as far as I understood)
- I could not find an analysis of the amortized running time. Is the amortized running time per update polylog? Update: the authors pointed our convincingly that their approach is almost optimal.
- the experimental evaluation could be improved. The authors considered only the first 5000 points for each dataset, claiming that there should not be much difference when processing the whole dataset. It is not clear whether this is the case.
- the experimental evaluation only considers the sliding window model. It would be more interesting to consider also updates that do not fit in the sliding window model, as they can be handled by the algorithm.

---

> ### Author Response · Authors · 2022-08-02
> **Addressing the comments/concerns**
>
> We thank the reviewer for their time and effort..
>
> ========== On comment on approximation ratio ==========
>
> On the constant approximation ratio. We note that we did not try to optimize the constant in the theoretical part of the work. Although we believe, it is possible to significantly reduce both the base of the exponent and the exponent itself. In fact, we did that in the empirical evaluation of the algorithm, where the NICECLUSTERING(µ = 3, ϵ = 0.05) variant performs almost identically to GreedyOff (which is known to provide an approximation guarantee of less than 2, see [1])
>
> ========== Comment on running time ==========
>
> [The answer is the same as for Reviewer GbJJ]
>
>
> The amortized update time of our algorithm is in fact sublinear in the size of the input instance. The size of the instance is $\Theta(mn)$, while our amortized update time is $\tilde{O}(m)$. Observe that it takes $\Omega(m)$ time to even specify an update, because in a general metric space an arriving client needs to specify its distances to all facilities. Hence, our update time is optimum, up to a polylogarithmic factor.
>
> We now compare our result with that of GKLX. Let us recall some facts from Section 2.1 of the GKLX paper (conference version). First, the total input size is $\Theta(n|F|)$  (in our paper $|F|=m$), which means that one cannot obtain a total update time of $o(nm)$ without further assumptions. GKLX makes the assumption that each client reports its nearest facility upon arrival, which avoids the $\Omega(mn)$ lower bound on the total update time. Thus, the model of GKLX is more restrictive than ours, since we make no such assumptions. GKLX further assumes that a client is always going to be co-located with some facility, which enables them to work with a HST embedding over the set of facilities (which is known in advance). Note that the moment an algorithm works with an HST embedding, it loses a logarithmic factor in the approximation ratio.
>
> Finally, if $n$ (the number of clients) is polynomially smaller than $m$ (the number of facilities), then the total update time of GKLX is polynomially larger than ours as it requires $O(m^2 \log m)$ preprocessing time.
>
>
> ========== Comment on experiments  ==========
>
> Regarding the comment on considering only the first 5000 points for each dataset, we will add an experiment showcasing this on at least one dataset.
>
> Regarding the consideration of different update sequences, we are not aware of openly accessible datasets with arrival/departure timestamps on points with embeddings attached to them. Hence, any assumption on the arrival sequence/departure is artificial. We chose this model, following the methodology of other experimental studies. If there exists another arguably realistic way of imposing arrival/departure sequences we would be happy to consider this in the final version of our paper.
>
> ========== Answering the main question ==========
>
> Regarding the question on whether the amortized running time of the main algorithm is polylog per update, the answer is no, it is bounded by $\tilde{O}(m)$. As we discussed above, this is near optimal (i.e., near-linear in the size of the input).
>
>
> ========== Citation of the answer ==========
>
> [1] Kamal Jain, Mohammad Mahdian, Evangelos Markakis, Amin Saberi, and Vijay V Vazirani. Greedy facility location algorithms analyzed using dual fitting with factor-revealing lp. Journal of the ACM (JACM), 50(6):795–824, 2003.

---

> > ### Comment · Reviewer_d29J · 2022-08-08
> > **On polylog update time**
> >
> > I would like to thank the authors for their answers, they address in a satisfactory way most of my questions. I still have one comment about the update time. The following two papers for k-center clustering have polylog update time (when the aspect ratio is polynomial):
> >
> > https://drops.dagstuhl.de/opus/volltexte/2016/6340/pdf/LIPIcs-ICALP-2016-19.pdf
> > https://arxiv.org/pdf/2112.07050.pdf
> >
> > Using the argument used by the authors: the input for k-center clustering in general metric is in $\Theta(n^2)$, hence, the update time should be $\Omega(n)$, which is clearly not the case. Could you please clarify what makes you clam that $\Omega(m)$ time is needed, in view of the above two papers? Thanks!

---

> > > ### Author Response · Authors · 2022-08-08
> > > **Re: On polylog update time**
> > >
> > > We thank the reviewer for their additional effort and time, and for taking into account our response.
> > >
> > > As we mentioned in our previous reply, to beat the $\Omega(m)$ update time (which is needed to describe the distances from the new point to all facilities), one needs to introduce some assumptions to avoid paying this cost.
> > >
> > > In the two papers mentioned by the reviewer, the authors make the assumption that they can query efficiently for the distance between any two arbitrary points in the metric space. (See in the preliminaries of the first paper, and the footnote 5 of page 13 and the sentences before in the second paper). The simplest way to do this, is to assume that the algorithm has access to an $n \times n$ matrix, storing the distances between any pair of points. Notice, however, that in the dynamic setting this requires $\Omega(n)$ (or in our case $\Omega(m)$) time to even update the entries of the matrix following the arrival of a point / client (if one does not have prior knowledge of the entire metric, which is not assumed in the general case). Even in scenarios where the distance oracle does not explicitly maintain the distance matrix, it would still need to read the input (that is, the distances from the point inserted to all other points of interest). Notice that these papers also do not account for the space complexity implied by the implementation of the assumed distance oracle, which could be as high as $\Omega(n^2)$. In our paper we assume no distance oracle, and rather directly read the input and store all required distances.
> > >
> > > In addition, one can consider a model where we disregard the space complexity (of storing the distance matrix) and measure our update time only in terms of the number of queries made to the distance oracle. Even in this model, the second paper obtains an upper bound of $\tilde{O}(k)$, which is parameterized by the output size (since an optimal solution opens k centers), and complements it with a matching lower bound. An analogous upper bound in our setting would give us an amortized update time of $\tilde{O}(\lambda)$, where $\lambda$ is the current number of facilities that are opened by our solution. We leave it as an interesting open question to decide whether one can design such an output sensitive algorithm. However, note that $\lambda$ can be as large as $m$, and if we care about a bound that is not output sensitive then this leads us back to our current bound of $\tilde{O}(m)$.
> > >
> > > Further, we note that if it was indeed possible to get a $O(1)$ approximate dynamic facility location algorithm with polylog amortized update time, then it would immediately imply the existence of a static algorithm for facility location with approximation ratio of $O(1)$ and a running time of $\tilde{O}(n)$: Simply insert all the clients one at a time and let the dynamic algorithm handle those insertions. Finally, output the solution after all insertions have been handled. To the best of our knowledge, there does not exist any such static algorithm for facility location.
> > >
> > > Finally, we remind that the update time is one of the measures that we aim to optimize. We also also prove a bounded recourse over the whole sequence of updates. Efficient dynamic algorithms with bounded recourse received increased attention (e.g., [1,2,3,4,5] just to name a few) and it is considered to be an important metric. Hence obtaining polylog amortized recourse for a fundamental clustering problem is an important contribution in its own right.
> > >
> > > ========== Citation of the answer ==========
> > >
> > > [1] Silvio Lattanzi, Sergei Vassilvitskii. Consistent k-Clustering. ICML 2017: 1975-1984.
> > >
> > > [2]  Anupam Gupta, Ravishankar Krishnaswamy, Amit Kumar, and Debmalya Panigrahi. Online and dynamic algorithms for set cover. In Proceedings of the 49th Annual ACM SIGACT Symposium on Theory of Computing, pages 537–550, 2017.
> > >
> > > [3] TH Hubert Chan, Arnaud Guerqin, and Mauro Sozio. Fully dynamic k-center clustering. In Proceedings of the 2018 World Wide Web Conference, pages 579–587, 2018.
> > >
> > > [4] Vincent Cohen-Addad, Niklas Oskar D Hjuler, Nikos Parotsidis, David Saulpic, and Chris Schwiegelshohn. Fully dynamic consistent facility location. Advances in Neural Information Processing Systems, 32, 2019.
> > >
> > > [5]  Hendrik Fichtenberger, Silvio Lattanzi, Ashkan Norouzi-Fard, and Ola Svensson. Consistent k clustering for general metrics. In Proceedings of the 2021 ACM-SIAM Symposium on Discrete Algorithms (SODA), pages 2660–2678. SIAM, 2021.

---

> > > > ### Comment · Reviewer_d29J · 2022-08-08
> > > > **Thanks!**
> > > >
> > > > I would like to thank the authors for their time and their detailed explanation, it makes sense to me. I will raise my score!

---

> > > > > ### Author Response · Authors · 2022-08-08
> > > > > **Thank you!**
> > > > >
> > > > > We thank again the reviewer for all their invested time and their thorough review. We appreciate that the discussion helped in clarifying and alleviating their main concerns.
> > > > >
> > > > > We will summarise the main points of the discussion in the next version of our paper.

---

### Official Review · Reviewer_GbJJ · 2022-07-11

**Rating:** 6
**Confidence:** 5
**Soundness:** 3 good
**Presentation:** 3 good
**Contribution:** 3 good

**Summary:**

The paper studies fully dynamic facility location for general metric spaces, where update operations are allowed to insert or delete clients. The main result is a dynamic algorithm which maintains a O(1)-approximate solution using O(log m) amortized recourse. Here, m is the number of facilities and the recourse is the number of clients and facilities that are reassigned after each update. The update time of the algorithm is O(m polylog(m)).

The approximation ratio improves upon a paper by Guo et al. (APPROX/RANDOM'20) which maintained a O(log m)-approximate solution with recourse and update time O(log m). The recourse of the new and the old result is the same, but the Guo et al. paper achieves a faster guaranteed update time.

The bound on the approximation ratio is obtained by defining a set of invariants and showing that when the invariants are satisfied, the maintained solution is "close" to the solution of an offline greedy algorithm which obtains a O(1)-approximation. It is then shown how these invariants can be maintained dynamically. The recourse bound is obtained using a technique introduced by Gupta et al. (STOC'17).

**Questions:**

My main question is how it is checked whether Invariant 4 is satisfied or not. This is not explained in the main text and I was not sure how to do this this efficiently. I looked at the corresponding section in the supplementary material, which provides a routine for finding blocking clusters. However, it is not immediately clear to me why this routine captures *all* possible blocking clusters (if there exists one). A proof for this might be interesting.

**Limitations:**

The authors have adequately addressed the limitations and potential negative societal impact of their work.

**Strengths And Weaknesses:**

Strengths:

(S1) Facility location is a fundamental problem and dynamic variants of it are highly interesting.

(S2) The theoretical results are interesting and non-trivial to obtain.


Weaknesses:

(W1) The main weakness of the theoretical result is certainly that only a bound on the recourse is obtained, while the bounds on space usage and update time are not very impressive. More concretely, in the streaming community one typically wishes for sublinear space usage and in the dynamic community one aims for sublinear update times; neither of this is achieved here.

*Update:* In the rebuttal, the authors pointed out that since a general metric space is considered, even specifying a single client insertion takes time $\Omega(m)$. Hence, an update time of $\tilde{O}(m)$ is somewhat inevitable. I still believe that this is property of the model is somewhat undesirable from a dynamic algorithms point of view, but I will not hold it against the paper.

(W2) The experimental evaluation is missing details and does not fully allow me to judge the quality of the results. For example, in Figure 3 (left), the line for GKLX is missing even though in the text (Lines 367--373) it is talked about it. It also seems like the proposed method only achieves factor 2.5 better approximation ratios than GKLX, while requiring much more recourse and update times. This certainly makes the result less interesting.

(W3) The submission contains quite a lot of typos.


Minor comments:
- Line 26: The word "studied" is missing.
- Line 26: "for for"
- Throughout the paper, the citation style is slightly confusing because it just lists author names but they are not used in the text (e.g., Line 47).
- Line 103: |C| is not defined, I presume it refers to |A|?
- I found the definition of ordinary and critical clusters a bit confusing and only understood it after I kept on reading.
- Lines 129--135: This part was hard to understand because nice clustering were not introduced yet.
- Line 136: Point out that C is not necessarily from \mathcal{C} but could be any possible cluster.
- Line 159: I find this reference to the relaxed greedy algorithm a bit confusing because this algorithm has not been formally defined.
- I find the size of the instances in the experiments a bit small. Having only 1000 points seems a bit unrealistic.
- Experiments: I would appreciate a clearer discussion about the tradeoff between better approximation and larger running times.
- Experiments: Since the algorithm works for general metric spaces, why were only L_2-distances considered?

---

> ### Author Response · Authors · 2022-08-02
> **Addressing the concerns/comments**
>
> We thank the reviewer for their time and effort.
>
> = W1 =
>
> The amortized update time of our algorithm is in fact sublinear in the size of the input instance. The size of the instance is $\Theta(mn)$, while our amortized update time is $\tilde{O}(m)$. Observe that it takes $\Omega(m)$ time to even specify an update, because in a general metric space an arriving client needs to specify its distances to all facilities. Hence, our update time is optimum, up to a polylogarithmic factor.
>
> We now compare our result with that of GKLX. Let us recall some facts from Section 2.1 of the GKLX paper (conference version). First, the total input size is $\Theta(n|F|)$  (in our paper $|F|=m$), which means that one cannot obtain a total update time of $o(nm)$ without further assumptions. GKLX makes the assumption that each client reports its nearest facility upon arrival, which avoids the $\Omega(mn)$ lower bound on the total update time. Thus, the model of GKLX is more restrictive than ours, since we make no such assumptions. GKLX further assumes that a client is always going to be co-located with some facility, which enables them to work with a HST embedding over the set of facilities (which is known in advance). Note that the moment an algorithm works with an HST embedding, it loses a logarithmic factor in the approximation ratio.
>
> Finally, if $n$ (the number of clients) is polynomially smaller than $m$ (the number of facilities), then the total update time of GKLX is polynomially larger than ours as it requires $O(m^2 \log m)$ preprocessing time.
>
> = W2 =
>
> Regarding the comment of the missing GKLX line in Figure 3 (left), the caption of Figure 3 states "Missing lines imply 0 recourse throughout.", which applies to GKLX. While GKLX incurs no recourse on this instance, it loses in the solution quality.
>
> On the comment that "only achieves factor 2.5 better approximation ratios than GKLX", we believe that a factor 2.5 reduction in the objective function can have a great impact on the application at stake, e.g., if the application models the total transportation cost of a business. Nonetheless, our algorithm is the first efficient dynamic algorithm to provably maintain a O(1)-approximate solution on any input.
>
> As a side note, we are not aware of an empirical evaluation of GKLX, and we consider this as one of our contributions.
>
> = W3 =
>
> We will fix all typos in the next version of our paper.
>
> = Minor comments =
>
> Comment: "Line 103…"
>
> Reply: C is defined in Line 102: "The average cost of a cluster C…"
>
> C: "I found the definition of ordinary…"
>
> R: We will address this in the next version of our paper.
>
> C: "Lines 129--135…"
>
> R: In Line 122 we informally define nice clustering: "The output of this relaxed greedy algorithm corresponds to what we call a nice clustering"
>
> C: "Line 136…"
>
> R: We point this out in the same line: "... identify a cluster C = (i, A) (not necessarily part of \mathcal{C})"
>
> C: "Line 159…"
>
> R: The algorithm is described in the "Our techniques" paragraph.
>
> C: "I find the size…"
>
> R: The main point of the experiments is to showcase the merits and limitations of our algorithm. We believe these experiments adequately communicate our message.
>
> C: "Experiments: I would appreciate…"
>
> R: In the paragraph "Conclusion of Experiments" we discuss the type of applications where our algorithm would be the best approach.
>
> C: "Experiments: Since the algorithm…"
>
> R: This is an arbitrary distance function that gives a good distribution of pairwise distances.
>
> = Question =
>
> Below, we discuss how we can efficiently check whether or not Invariant 4 is satisfied.
>
> For every facility $i$ and level $k$, let $S(i, k)$ denote the sequence of clients that are at a level $> k$, in increasing order of their distances from facility $i$. The key observation is this: If the set of blocking clusters at level $k$ with facility $i$ is nonempty, then there must exist a blocking cluster (with the same facility and at the same level) whose clients form a prefix of $S(i, k)$. This has the following implication. Suppose that we explicitly maintain these sequences $\{S(i, k)\}$. Then given a specific pair $(i, k)$, we can determine in only $\tilde{O}(1)$ time (via a simple binary search and some standard data structures) whether or not there is a blocking cluster involving facility $i$ at level $k$. Finally, note that we can indeed maintain the sequences $\{ S(i, k) \}$ by paying a factor $\tilde{O}(m)$ overhead in our update time. This is because whenever a client changes its level (or gets inserted / deleted) we need to modify at most $mL = \tilde{O}(m)$ of these sequences $\{ S(i, k) \}$, where $L$ is the number of levels.
>
> In the supplementary material, we present a more fine tuned data structure than the one described above, where we discretize the distances in powers of $(1+\epsilon)$. However, the main idea behind the data structure is the same as described above. We will explain this in more detail in the final version of the paper.

---

> > ### Comment · Reviewer_GbJJ · 2022-08-04
> > **Thank You!**
> >
> > I thank the authors for their clarifications and explanations, in particular regarding W1 and my question about Invariant 4. I will increase my score accordingly.
> >
> > Minor comments:
> > - Line 103: I was not referring to C but to |C|. Since C=(i,A) is a tuple, it is not immediately clear what |C| is supposed to be. I guess that |C| := |A| but that should be said somewhere.
> > - Regarding Figure 3, I understand the performance of GKLX now but I still find that not adding this line to the figure is confusing and a bit misleading.

---

> > > ### Author Response · Authors · 2022-08-08
> > > **To incorporate clarifications in the paper.**
> > >
> > > We thank the reviewer for their additional effort put into taking into account our response. We appreciate that our explanation helped in clarifying and alleviating the main concerns of the reviewer.
> > >
> > > We will add this discussion, as well as clarifying the remaining minor comments, in the next version of our paper.

---

### Official Review · Reviewer_f1Wt · 2022-07-11

**Rating:** 7
**Confidence:** 4
**Soundness:** 3 good
**Presentation:** 3 good
**Contribution:** 3 good

**Summary:**

Dynamic Facility Location problem (with client insertions and deletions) is being considered in which modifying the solution involves additional cost (recourse). The authors provide an algorithm that maintains a constant factor approximation solution to the current instance
with the additional guaranty that the amortized recourse is polylogarithmic in the instance size.

Technically, the result is obtained by an adaptation of the greedy JMS algorithm for the static case. The solution maintained by the algorithm is formally proved to be close to what the greedy algorithm would produce for the current instance, which suffice to argue for it being a constant factor approximate solution.

**Questions:**

Minor comments:
- line 26 "been extensively" -> "been studied extensively"
- l. 26 "for for"
- l. 209 "repeated checks" -> perhaps "repeatedly checks"?
- l. 254 and l. 256 "total units of ..." -> either " total ..." or "total number of units of ..."

**Limitations:**

The experimental part dresses some practical limitations adequately.

**Strengths And Weaknesses:**

Strengths:
* it is one of the central optimization problems (not a very special case of it)
* the provided solution is elegant
* the obtained result appears to be relevant

Weaknesses:
* I find the notion of "ordinary clusters" somewhat annoying, these are just "one client connected to one facility" in the end.
  It appears to be there just to enable using term "clusters" for both the real clusters called "critical clusters" in the paper, and these client-facility pairs called "ordinary clusters" in the paper.

---

> ### Author Response · Authors · 2022-08-02
> **Acknowledging the suggestion**
>
> We thank the reviewer for their time and effort.
>
> We will come up with a more appropriate name for the concept of ordinary clusters in the final version of our paper.

---

### Official Review · Reviewer_6GBX · 2022-07-12

**Rating:** 7
**Confidence:** 3
**Soundness:** 4 excellent
**Presentation:** 4 excellent
**Contribution:** 3 good

**Summary:**

The paper considers the facility location problem in the fully dynamic setting, where clients can come and go. The goal is to maintain a near-optimal solution and simultaneously minimize the recourse, namely the number of facility openings/closings $+$ reassignments of clients to centers. There is a caveat: the algorithm in this paper only works when the ratio of maximum distance / opening cost to minimum distance / opening cost is bounded polynomially in the number of candidate facility locations $m$. In this setting, the paper gives a dynamic algorithm that maintains a constant approximation guarantee, and uses $O(\log m)$ amortized recourse per change. The main idea is to relax the greedy algorithm which picks repeatedly a cluster (facility $+$ connected clients) of minimum average cost, then show how to maintain such a solution dynamically with small recourse. There are some experimental results which I don't know how to evaluate.

**Questions:**

Are there any lower bounds on the achievable recourse?

**Limitations:**

Irrelevant.

**Strengths And Weaknesses:**

The dynamic setting is well-motivated and the solution is quite interesting.

---

> ### Author Response · Authors · 2022-08-02
> **"Answering the question"**
>
> We thank the reviewer for their time and effort.
>
> The current lower bound on the total recourse is $\Omega(n)$ [1], where $n$ is the number of clients, which holds even in the case of uniform facility opening costs. It is an open research problem whether there exists an algorithm that can match this lower bound.
>
>
> [1] Vincent Cohen-Addad, Niklas Oskar D Hjuler, Nikos Parotsidis, David Saulpic, and Chris Schwiegelshohn. Fully dynamic consistent facility location. Advances in Neural Information Processing Systems, 32, 2019.

---

> > ### Comment · Reviewer_6GBX · 2022-08-08
> > **Ack**
> >
> > Thank you for answering my question.

---

### Meta-Review · Area_Chair_vn9v · 2022-08-27

**Recommendation:** Accept
**Confidence:** Certain

**Metareview:**

The paper considers facility location problem in a well-motivated fully-dynamic setting where clients can arrive and depart. The goal is to maintain a near-optimal solution and simultaneously minimize the amount of "recourse" (the number of facility openings/closings and reassignments of clients to centers). However, this algorithm only works when the ratio of maximum distance / opening cost to minimum distance / opening cost is bounded polynomially in the number m of given facility locations; the paper presents an algorithm that maintains a constant-factor approximation guarantee while using an O(log m)--amount of amortized recourse per change. The idea is to relax the classical greedy algorithm of Jain-Mahdian-Saberi for the static case.which picks repeatedly a cluster---a facility along with the clients assigned to it---of minimum average cost, and to show how to maintain such a solution dynamically with small recourse.

The paper was generally strongly appreciated by the reviewers.

**Award:**

No

---

### Decision · Program_Chairs · 2022-09-14

Accept